

# Atmospheric observations of the water vapour continuum in the near-infrared windows between 2500-6600 cm[-1]

Jonathan Elsey[1], Marc. D. Coleman[2], Tom Gardiner[2], Kaah P. Menang[3], Keith P. Shine[1]

[1]Department of Meteorology, University of Reading, Reading, RG6 6BB, United Kingdom
[2]National Physical Laboratory, Teddington, London, TW11 0LW, United Kingdom
[3]National Department of Meteorology, Ministry of Transport, Yaoundé, Cameroon

*Correspondence to*: Jonathan Elsey (jon.elsey@reading.ac.uk)

**Abstract.** Water vapour continuum absorption is potentially important for both closure of the Earth's energy budget and remote sensing applications. Currently, there are significant uncertainties in its characteristics in the near-infrared atmospheric

windows at 2.1 and 1.6 μm. There have been several attempts to measure the continuum in the laboratory; not only are there significant differences amongst these measurements but there are also difficulties in extrapolating the laboratory data taken at room temperature and above to temperatures more widely relevant to the atmosphere. Validation is therefore required using field observations of the real atmosphere. There are currently no published observations in atmospheric conditions with enough water vapour to detect a continuum signal within these windows, or where the self-continuum component is significant.  We

present observations of the near-infrared water vapour continuum from Camborne, UK at sea level using a sun-pointing, radiometrically-calibrated Fourier transform spectrometer in the window regions between 2000-10000 cm[-1]. Analysis of this data is challenging, particularly because of the need to remove aerosol extinction, and the large uncertainties associated with such field measurements. Nevertheless, we present data that is consistent with recent laboratory datasets in the 4 and 2.1 μm windows (when extrapolated to atmospheric temperatures). These results indicate that the most recent revision (3.2) of the

MT_CKD foreign continuum, versions of which are widely used in atmospheric radiation models, requires strengthening by a factor of ~5 in the centre of the 2.1 μm window. In the higher-wavenumber window at 1.6 μm, our estimated self and foreign continua are significantly stronger than MT_CKD. The possible contribution of the self and foreign continua to our derived total continuum optical depth is estimated by using laboratory or MT_CKD values of one, to estimate the other. The obtained self-continuum shows some consistency with temperature-extrapolated laboratory data in the centres of the 4 and 2.1 μm

windows. The 1.6 μm region is more sensitive to atmospheric aerosol and continuum retrievals and therefore more uncertain than the more robust results at 2.1 and 4 μm. We highlight the difficulties in observing the atmospheric continuum and make the case for additional measurements in both the laboratory and field and discuss the requirements for any future field campaign.



## 1. Introduction

### 1.1: Background

The near-infrared spectrum (defined here in wavenumber space as 2000-10000 cm$^{-1}$) is characterised by its spectral band-window structure, where parts of the spectrum are completely opaque to radiation and others are mostly transparent over

typical (clear-sky) atmospheric paths. Within this spectral region, in addition to the many discrete spectral lines of various gases, there is additional absorption due to the water vapour continuum absorption (henceforth simply continuum), a smoothly varying (with wavenumber) component of the total absorption which underlies this band-window structure. The cause of this continuum is not known but is postulated to be due to a combination of far-wing broadening, e.g. by collisional effects, and absorption due to water dimers (bound or quasi-bound complexes of two water vapour molecules), as discussed in e.g. Shine

et al., (2012). The continuum is normally broken down into two components; a self-continuum component that depends on the square of the vapour pressure, and a foreign continuum component that depends linearly on vapour pressure and the pressure of the ambient air. The foreign continuum is observed to have a very weak temperature dependence (Ptashnik et al., 2012), while the self-continuum has a negative exponential temperature dependence (Mondelain et al., 2014; Ptashnik et al., 2011a). The temperature dependence of the self-continuum is broadly consistent with a dimer-like theory, but this has not been verified

due to the difficulty of performing *ab initio* calculations of the water dimer spectrum, and the strength of the temperature dependence varies amongst different sets of measurements and may depend on wavenumber (e.g. Ptashnik et al., 2019)).

Since the continuum absorbs radiation (particularly in the atmospheric windows) which would otherwise penetrate further into the atmosphere or reach the surface, it influences the surface-atmosphere partitioning of energy and is therefore important for

understanding the global energy budget. In the more transparent window regions, most of the continuum absorption occurs in the troposphere where water vapour is more abundant and has a potential influence on the hydrological cycle. The continuum contribution to climate feedbacks could also be enhanced in a warming climate via the water vapour feedback; the strongly absorbing water vapour bands are already close to saturation, meaning that the window regions, in which the continuum is comparatively more important, could contribute more to the change in absorption in a warming climate. For example, Rädel

et al. (2015) found that the near-IR continuum contributes ~10-20% of the total water vapour shortwave feedback in the scenario with a 33% increase in water vapour, depending on whether a weaker or a stronger continuum is used. The continuum also impacts upon remote sensing of the Earth's atmosphere and surface. Some remote sensing platforms e.g. the Orbiting Carbon Observatory 2 (OCO-2) (Oyafuso et al., 2017) have channels observing in the 2.1 and 1.6 µm (~4000 and 6300 cm$^{-1}$ respectively) windows, as does the MODIS satellite (Platnick et al., 2017), which is used to retrieve gas concentrations, cloud

properties, surface albedo and aerosol optical depth.

The strength of the near-infrared continuum is uncertain, particularly in the 2.1 and 1.6 µm windows. There have been relatively few attempts to measure the self-continuum in the laboratory, with observed absorption coefficients that differ significantly

(e.g. Shine et al., 2016) in the centres of these windows at room temperature. Measuring the continuum in the laboratory is problematic in some ways, due to the need to extrapolate in temperature and pressure to conditions present in the atmosphere (which are frequently below room temperature). The weak absorption strength of the continuum in the windows makes it difficult to measure at typical tropospheric temperatures (~280 K) without long path lengths (such as that from the top of

atmosphere (TOA) to the surface) which are difficult to attain in a laboratory. These issues can be mitigated using certain high-precision techniques (e.g. cavity ring-down spectroscopy (CRDS)), at the cost of wide spectral coverage; however, while CRDS measurements exist at room temperature, at there are none reported in the literature at the lower temperatures considered here. Additionally, the weak (and featureless) absorption means that the measurements are very sensitive to the experimental conditions, such as the baseline stability of the spectrometer when using Fourier transform spectroscopy (FTS) techniques (e.g.

Ptashnik et al., 2015).

The continuum is parameterised in most radiative transfer codes used in models and remote sensing by the MT_CKD (Mlawer-Tobin_Clough-Kneizys-Davies) model (Mlawer et al., 2012); in many cases they use either version 2.5 or version 3.2 (Mlawer et al., 2019). MT_CKD is a semi-empirical model. In the window regions, the MT_CKD continuum mostly originates from

the adjustment of the water vapour lineshape using a $\chi$-factor derived primarily from measurements at wavenumbers in the mid and far-infrared (< 2000 cm$^{-1}$), with additional empirical adjustments. It is not an *ab initio* calculation, and uses selected observations to adjust its continuum strength. Any such adjustment should therefore consider the uncertainty and differences in the available measurements. A particularly important aspect is the temperature dependence; atmospheric radiative transfer models generally use the MT_CKD formulation to extrapolate the self-continuum absorption to temperatures at which there

are no laboratory measurements.

Measurements of the continuum in the atmosphere are therefore necessary to supplement laboratory measurements. While field measurements present their own issues, explained more in Sections 3 and 5, they provide data with which to test the experimentally-implied temperature dependence, as well as that of MT_CKD. Ideally, a combination of field and laboratory

measurements would converge on a set of continua at different temperatures and pressures that could be included into spectroscopic databases such as HITRAN (Gordon et al., 2017), or at least provide a set of robust values (with agreement within the uncertainties) that can be used to adjust MT_CKD.

In this work, we present the first reported derivation of the near-IR atmospheric continuum in the 4, 2.1 and 1.6 μm windows

at mean sea level with a well-constrained uncertainty budget, and the first to be derived using a radiometrically calibrated spectrometer. These measurements were made during the CAVIAR (Continuum Absorption at Visible and Infrared wavelengths and its Atmospheric Relevance) field campaign in Camborne, Cornwall, UK in August-September 2008 (Gardiner et al., 2012). Since these measurements are at mean sea-level, it has been estimated that the continuum absorption will be roughly evenly split between the self and foreign continua at 1.6 μm, and ~70:30% in the 2.1 μm window based on laboratory





measurements (as calculated in Ptashnik et al., 2012). Additionally, observing at sea-level allows us to measure the continuum within the windows, as the expected continuum contribution is above the signal-to-noise of our spectrometer (see Section 2). These conditions set our results apart from those of Reichert and Sussmann, (2016), who used an FTS at a high altitude site to measure the continuum. This allowed observations of the continuum within the bands but restricted the ability to detect it

within the windows. Additionally, our measurements are radiometrically calibrated and traceable to SI (*Système international d'unités,* BIPM, 2006); this allows us to obtain the top-of-atmosphere solar spectral irradiance (SSI) directly (Elsey et al., 2017; Menang et al., 2013), which is itself uncertain to ~8% in the 4000-7000 cm$^{-1}$ region.

**1.2: Atmospheric observations of the near-IR continuum**

This Section discusses the current literature in terms of field measurements of the near-IR continuum. Reichert and Sussmann,

(2016), henceforth "Zugspitze", presented a continuum absorption obtained in atmospheric conditions at a high-altitude site at the Zugspitze in the German Alps. This used an FTS calibrated using a combination of Langley-derived TOA irradiance, a medium-temperature (~1970 K) blackbody and an assumed SSI from a radiative transfer model (Reichert et al., 2016). The high altitude allows for measurements of the continuum well into the main water vapour absorption bands and ostensibly allows for an upper limit to be set on the absorption in the windows. These are the most immediately comparable measurements

in the literature to the ones presented here. There are several key differences between the two field campaigns which makes them difficult to compare directly. The Zugspitze measurements were performed in conditions that had a significantly smaller water vapour path, meaning that observations of the continuum in the windows are extremely difficult, while allowing observations in the bands that sea-level observations are not capable of. Additionally, the higher altitude measurements are dominated by the foreign continuum due to the lower vapour pressures, whereas the sea-level observations are more of a

mixture of foreign and self-continua. The higher altitude measurements are above the atmospheric boundary layer, mitigating the effect of aerosol extinction which is a significant problem for sea-level observations. To obtain a long enough path length to mitigate the lack of water vapour, the Zugspitze measurements were taken at large airmass factors (~6 airmasses). This may be problematic however since a) the effects of atmospheric refraction are more pronounced, and b) extrapolating from high airmass to zero airmass using the Langley method increases the effect of the uncertainty in the individual measurements, since

these primarily use the closure method and are therefore reliant on their calibration to a prescribed SSI.

These factors mean that Zugspitze observations are available in the 2.1 μm window and within several of the adjacent water vapour bands, but values are not presented in the 1.6 μm window (many of these are in fact negative). Due to the large uncertainties, these are seemingly consistent with both MT_CKD and contemporary laboratory measurements of the foreign

continuum (see Section 4.2), despite the considerable differences between these datasets. These will be examined in more detail in Section 4. Nevertheless, it should be emphasised that these measurements are a significant advance in our understanding of the in-band continuum. Additionally, as understanding of the near-IR SSI is improved, the calibration used in the Zugspitze measurements could be used to measure the continuum without the need for an expensive and time-consuming





blackbody calibration, which would allow for measurements in a wider variety of conditions. This would both help validate radiative transfer models and allow for separation of the foreign and self-continuum contributions in atmospheric conditions; this task is extremely challenging to do with a single field campaign at one location if only modest changes in water vapour column occur.

## 2. Methods and experimental setup

This work builds upon the work of Tallis et al., (2011), Menang et al., (2013), and Elsey et al., (2017). These all used observations obtained using an absolutely-calibrated ground-based sun-pointing Fourier transform spectrometer (Gardiner et al., 2012) set up at a field site in Camborne, Cornwall, UK (50.218ºN, 5.327ºE). Those papers focused on water vapour spectral lines and SSI respectively. Gardiner et al., (2012) presents the calibration procedure and FTS setup in detail. The spectrometer measures the centre of the solar disk (using dedicated solar tracker optics) in the range 2000-10000 cm$^{-1}$, with a spectral resolution of 0.03 cm$^{-1}$. The FTS is radiometrically calibrated, with traceability to SI via calibration to the 3000 K Ultra High Temperature Blackbody (UHTBB) at the UK National Physical Laboratory.

The total optical depth $\tau_{total}$ can be determined from the irradiance $I$ observed by the FTS at a given airmass factor $m = \sec(\theta)$ (with $\theta$ the solar zenith angle). This is done using measurements at a range of airmasses via the Langley method, or given a top-of-atmosphere irradiance $I_0$, the radiative closure method. Taking the logarithm of the Beer-Bouguer-Lambert law:

(1)  $\ln(I) = \ln(I_0) - m\tau_{total}$.

The radiative closure method is a simple inversion of this equation to solve for $\tau_{total}$. In this case, the SSI of Elsey et al., (2017) is used, since this is determined directly by the spectrometer used in this work. This does however introduce significant extra uncertainty, given the large uncertainty in the near-IR SSI, particularly in the lower-wavenumber windows.

The Langley method exploits the fact that Eq. (1) can be solved as a linear equation given observations at various airmasses, assuming that the optical depth does not vary significantly between these airmasses. This means that the aerosol optical depth ($\tau_{aerosol}$) needs to be measured at the same time as a spectrometer measurement and along the same atmospheric path, as does the integrated water vapour (IWV). It also means that measurements must be taken when there are no clouds present. $\tau_{aerosol}$ was measured using a handheld Microtops II sunphotometer (Solar Light Company, 2001) at 0.38, 0.44, 0.675, 0.936 and 1.02 µm. Integrated water vapour was measured using a HATPRO microwave radiometer (Rose and Czekala, 2009). The effects of clouds were minimised by visually checking for clouds at the time of measurement, and by using the variation in the observed voltage of the spectrometer detector to determine whether any sub-visible clouds or haze passed into the line-of-sight of the spectrometer during a measurement.


The continuum optical depth, $\tau_{cont}$ derived from the total optical depth $\tau_{total}$ obtained from the spectrometer measurements, can be characterised as:

$$(2) \qquad \tau_{cont} = \tau_{total} - \tau_{H2O_{lines}} - \tau_{other\_gases} - \tau_{Rayleigh} - \tau_{other} - \tau_{aerosol}.$$

The retrieval of the continuum mostly relies on accurate determination of the line-by-line absorption from water vapour and other gases, and aerosol extinction. Rayleigh scattering was modelled using the calculations of Bucholtz, (1995). It is mostly negligible in the near-IR windows (Elsey et al., 2017) and thus has minimal effect on the derived continuum. The line-by-line optical depth was determined using the Reference Forward Model (version 5.01, Dudhia, (2017)) and the HITRAN2016 spectroscopic database (Gordon et al.. 2017) and the Voigt lineshape cut off at 25 cm$^{-1}$ (with the line contribution at 25 cm$^{-1}$

subtracted at wavenumbers less than 25 cm$^{-1}$, as this is assumed to be part of the continuum following the MT_CKD definition). The atmospheric profiles were derived using co-located radiosonde ascents and checked using ECMWF and Met Office analysis data. To minimise the effect of solar lines, all regions within 0.1 cm$^{-1}$ of a solar line (as observed by Menang et al., (2013) and Elsey et al., (2017)) are filtered out. To minimise the effect of line shifting in the measurements or misattributed line positions in HITRAN, the observed continuum is smoothed over 15 cm$^{-1}$. This smoothing is suitable for observing the

continuum, as the continuum varies smoothly with wavenumber. This is necessary in particular due to the high spectral resolution of the measurements, and also filters out any high frequency noise within these observations that may not be accounted for otherwise. Regions with $\tau_{total}$ above 0.1 are also filtered out, to ensure that continuum derivation only takes place within microwindows, and in regions where the modelled spectral lines can be reasonably subtracted from the observed ones (where the absorption is not saturated).

Continuum absorption by other molecules ($N_2$, $O_2$, $O_3$ and $CO_2$, defined here as $\tau_{other}$) was obtained from MT_CKD3.2 (Mlawer et al., 2012; 2019). This non-water vapour continuum absorption is mostly important in the 1.25 μm window, where there is significant absorption due to a collision-induced oxygen band; however, this window is not the focus of discussion here. Figure 1 shows a schematic of how this information is put together to retrieve the continuum from the FTS measurements.





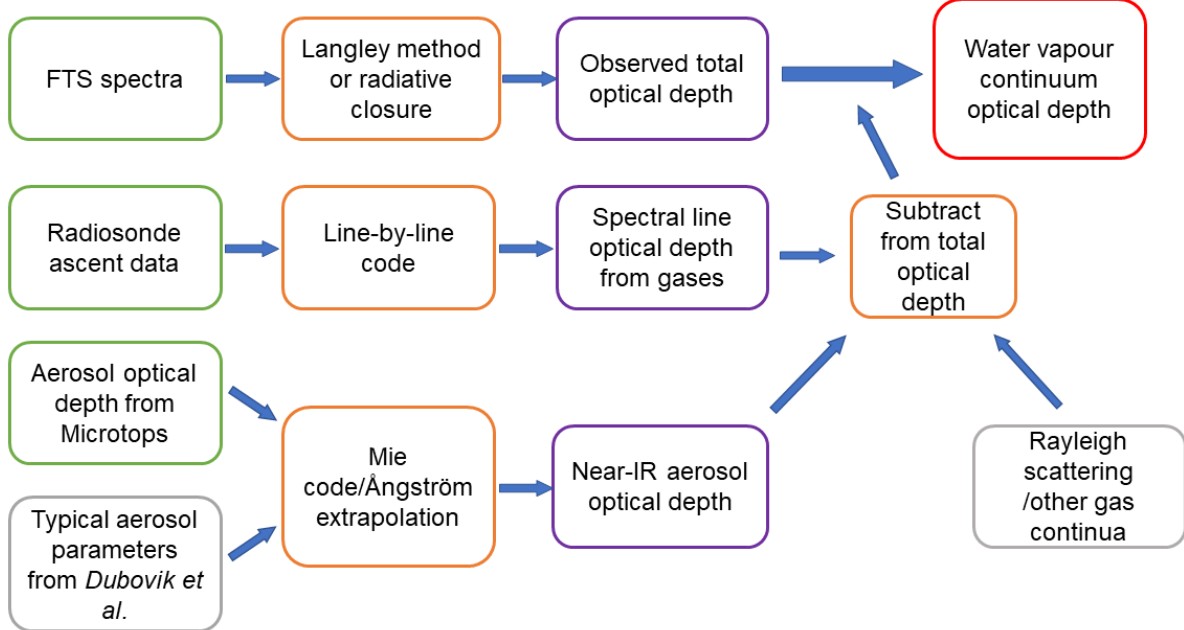

**Figure 1: Schematic of the derivation process of the water vapour continuum from the primary data (green), supplementary data (grey), computational methods (orange), intermediate outputs (purple) and final output (red).**

The best estimate of the continuum is from measurements made on 18 September 2008, with additional observations from other days. The IWV observed by the HATPRO microwave radiometer on 18 September was $16.25 \pm 0.49$ kg m$^{-2}$. The reliance on the observations of 18 September 2008 is due to the need to observe in clear skies, and to minimise the effects of atmospheric aerosol. 18 September 2008 had clear skies for most of the day, allowing observations at a wide range of airmasses for Langley

10  extrapolation. Additionally, the aerosol optical depth was significantly lower (observed via the sunphotometer) than the other days that fit this criterion. This is a significant issue for a continuum derivation; when deriving SSI a small absolute change in aerosol optical depth across the course of a day has a minimal effect on the y-intercept of the Langley plot, but the effect on the gradient (i.e. optical depth) is comparatively much larger. Constraining the aerosol change throughout the day is a significant challenge for such sea-level observations. Since the analysis is reliant mostly on one day of observations, and given

15  the large uncertainties, it is not possible to retrieve the self or foreign continua separately. Therefore, to compare with the laboratory measurements, an assumption needs to be made about the relative strength of either the foreign or self-continuum (see Section 4). Figure 2 shows four Langley plots from the 18 September 2008 data, at wavenumbers in the 4, 2.1, 1.6 and

1.02 μm windows. These plots demonstrate the quality of fit (and therefore the strong constraint on the observed total optical depth) we were able to obtain from the observations of 18 September 2008.

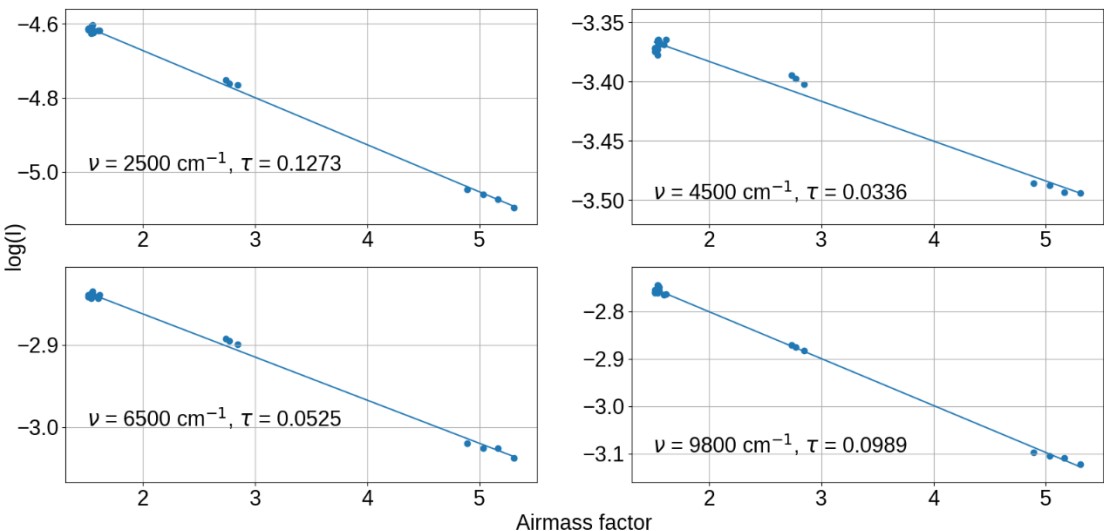

**Figure 2: Langley plots from selected wavenumbers in the 4, 2.1, 1.6 and 1.02 μm atmospheric windows, along with the total optical depth obtained at that wavenumber. Taken from observations of 18 Sept 2008.**

Figure 3 shows the derivation process in the 1.6 μm window, starting with the Langley-derived $\tau_{total}$ from the FTS observations (panel a), subtracting the line-by-line contributions (panel b), smoothing and subtracting Rayleigh scattering and other gaseous continua (panel c), and finally obtaining the water vapour continuum by subtracting aerosol extinction (panel d).



**Figure 3: Example derivation of the water vapour continuum optical depth $\tau_{cont}$ from the total optical depth $\tau_{total}$ (a) via subtraction of $\tau_{H2O\_lines}$ and $\tau_{other\_gases}$ (b), smoothing and subtraction of Rayleigh scattering $\tau_{Rayleigh}$ and continuum absorption by other gases $\tau_{other}$(c), and finally subtracting $\tau_{aerosol}$ to get the water vapour continuum optical depth (d).**

Figure 4 shows the minimum detectable optical depth capable of being observed by the FTS. This was calculated using the following method. For a series of repeated observations from the calibration campaign (measurements of the UHTBB, see Gardiner et al., (2012) for more details) the window regions (2500 – 2800; 4400 – 4800; 6000 – 6400; 7900 – 8400; 9200 –





10000 cm[-1]) were selected. In each window region, the mean signal level was calculated for each measurement. From this, the absolute difference between these levels and the mean level across *all* the measurements was obtained. The average difference gives a measure of the noise on this difference in each region. We then take an observation of the Sun (one used in the Langley analysis) and calculate the mean solar irradiance signal in each spectral window. The ratio of the offset noise to the solar signal

5    gives the fractional offset noise in each window which, using the small absorption approximation, is approximately the optical depth noise in that region. The minimum detectable offset is then assumed to be 3 times the optical depth noise. It is found that the minimum detectable optical depth in each of the atmospheric windows is typically 0.001, significantly below the derived continuum optical depth in most cases.

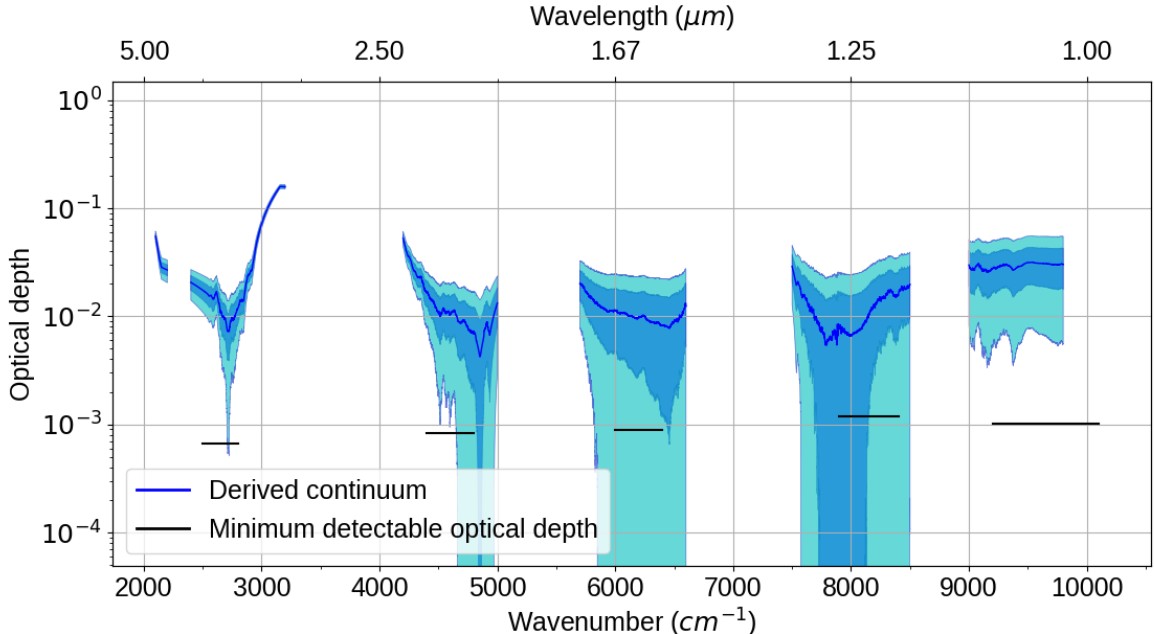

**Figure 4: Minimum detectable optical depth in the various atmospheric windows presented in this work (horizontal black line) against derived continuum optical depth from 18 September 2008 (blue line). The shadings indicate the $k = 1$ (blue) and $k = 2$ (cyan) uncertainty limits.**

15    The uncertainty budget on the continuum optical depth is obtained in a similar way to Elsey et al., (2017). The Monte Carlo method used there was extended to obtain the experimental uncertainty in the total optical depth. Uncertainty in the optical depth from the line-by-line model comes from sensitivity tests using the uncertainty limits in temperature, pressure and water vapour from the radiosonde. Due to the increased sensitivity to the atmospheric aerosol (when deriving continuum absorption rather than SSI), $\tau_{aerosol}$ was determined using the Microtops measurements and a Mie scattering code based on Wiscombe,

20    (1980), in addition to the Ångström exponent method described in Elsey et al. (2017). The Mie code was fed with a range of parameters for a comparable atmosphere obtained from Dubovik et al., (2002). This allowed us to test the range of validity of the Ångström exponent method, by using a physically-based wavelength dependence. The uncertainty budget was more





conservative than that of Elsey et al. (2017), since this was estimated using the Mie scattering calculations which were sensitive to various parameters (e.g. size distribution) which had large ranges in Dubovik et al., (2001). Figure 5 shows the optical depth and $k = 1$ (67 % confidence interval) uncertainties of the $\tau_{aerosol}$ used in this work, and the relative contribution this optical depth has to the combined continuum + aerosol optical depth.

**Figure 5: Panel a): Aerosol optical depth obtained from the Mie scattering calculations for 18 September 2008 with the Microtops $\tau_{aerosol}$ at 1 $\mu m$, along with the estimated $k= 1$ uncertainties (shaded region). Panel b): Relative contribution of the continuum and**
10 **aerosol in each of the near-infrared windows to the combination of the two ($\tau_{aerosol} + \tau_{continuum}$).**





An issue with our derivation of $\tau_{aerosol}$ is our inability to reconcile the observed variation in the sunphotometer $\tau_{aerosol}$ (+ $\tau_{cont}$ in the 1 μm channel, since this is not corrected for in the Microtops processing algorithm) on 18 September 2008 with the variation in the Langley-derived $\tau_{aerosol} + \tau_{cont}$ from the FTS. Figure 6 shows the time variation of the IWV and the continuum plus aerosol optical depth from the FTS and the sunphotometer. The FTS showed a consistent combined continuum

and aerosol optical depth throughout the day, while the Microtops showed a significant drop in aerosol optical depth over the course of the day. This is very unlikely due to the continuum, since the IWV observed by the HATPRO varied by only ~5% throughout the day, which would not be enough to cause such large changes. The surface temperature as observed by the radiosondes varied by less than 1 K throughout the period of measurement. Additionally, the Microtops does not contain a correction for the water continuum; if there was a significant change in continuum absorption then this again should be seen

in both the Microtops and FTS data.

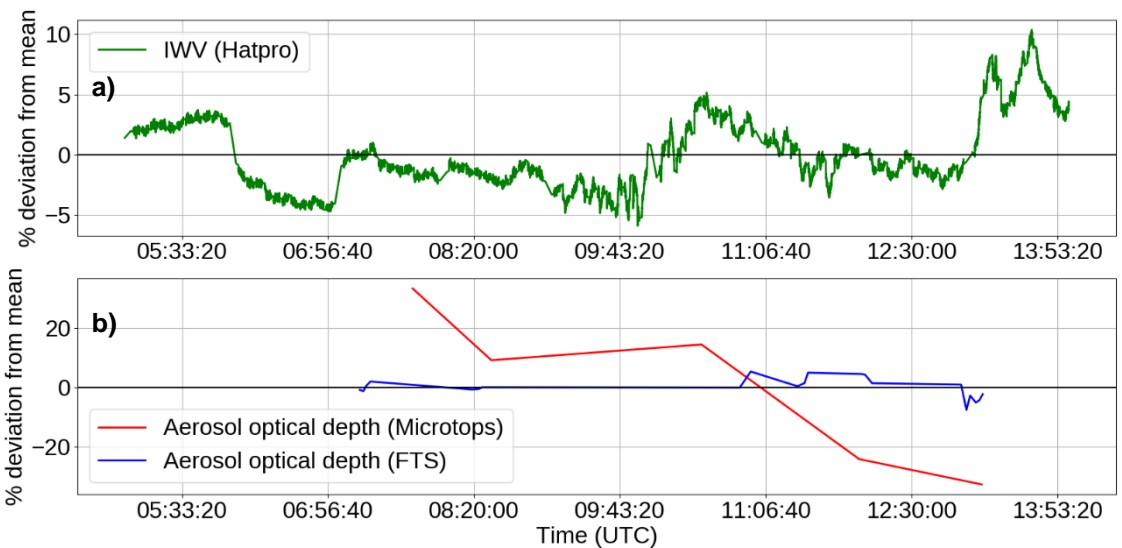

**Figure 6: Percentage variation across 18 September 2008 in integrated water vapour as measured by the HATPRO microwave radiometer (a) and aerosol (+ continuum) optical depth as measured by the Microtops sunphotometer and the FTS (b) in the 1.02**
**μm channel (9583 cm⁻¹ in FTS). The FTS-observed aerosol optical depth (+ water vapour continuum) is ~0.1 in this window, while the Microtops-observed aerosol optical depth is ~0.05.**

It is therefore unclear what is causing this discrepancy in the time-variation, but it may be due to uncertainties arising from the operation of the sunphotometer, or some systematic time-varying effect impacting the FTS measurements. For the continuum

derivation it was decided to use the day-average of the 18 September 2008 $\tau_{aerosol}$ measurements, with a corresponding increase in the uncertainty, since we could not determine which aerosol variation was more likely to be the true case.

In addition to the issue with the variation, there is also an unreconcilable difference between the optical depth observed by the FTS at 1 μm (~0.1) and that observed by the Microtops (0.03 – 0.08). Due to the small variation in IWV and temperature across the day, the larger signal observed by the FTS is extremely unlikely to be due to water vapour absorption. It is unclear why the FTS and the Microtops do not observe the same signal. If the effect were physical, one would expect the Microtops

and FTS to both observe it. While the variability in the Microtops is large, the absolute level of $\tau_{aerosol}$ is believed to be more reliable than that from the FTS, particularly given the consistency with shorter-wavelength Microtops measurements. This makes it a more reliable instrument for extrapolating optical depth to lower wavenumbers. It was postulated that the discrepancy may be due to a change in forward scattering with wavenumber and the differences between the field-of-view of the Microtops and the FTS, but this correction to $\tau_{aerosol}$ is less than 10% at all wavenumbers observed by the FTS (Box and

Deepak, 1979)

Another issue is the assumptions made regarding the mirror reflectivity correction. Since the mirrors were exposed to the elements, a correction is made to the observed irradiance based on observations of the mirrors prior to the field campaign, and subsequent measurements afterward using the National Reflectance Reflectometer (NRR) at NPL. However, the NRR

observations only cover the spectral region 4000-6600 cm$^{-1}$. The reflectance outside of these regions must be extrapolated based on the observations within this spectral region. It is for this reason that we have more confidence in the observations at these wavenumbers, and in the adjacent windows where the extrapolation takes place over fewer wavenumbers. There is significant uncertainty in the behaviour in the 1 μm window, where the mirror correction is extrapolated further, which may be in excess of the uncertainty estimate in Gardiner et al. (2012). The Supporting Information has more details on the possible

effect of this mirror extrapolation.

It was postulated that there could be significant uncertainty at higher wavenumbers ( > 7500 cm$^{-1}$) due to some uncertainty or systematic offset in the phase correction used in the OPUS software used to derive spectra from the FTS measurements (see Supporting Information). This was motivated by the observation of systematic changes in the FTS phase spectrum with respect

to time across 18 September 2008, that were particularly large at higher wavenumbers. It was found that uncertainties in this phase correction would have small effects at lower wavenumbers, but could significantly impact the observed optical depth at higher wavenumbers. However, we do not have a physical justification for why this may have been the case and cannot *ab initio* determine the magnitude of this uncertainty.

We believe that the combination of the above factors (mirrors, phase correction issues, larger aerosol effect) warrants significant caution being used when interpreting the results at wavenumbers beyond ~6700 cm$^{-1}$. The observed optical depth (see Section 3) is seemingly inconsistent with the (admittedly sparse) laboratory estimates or MT_CKD. Therefore, the apparently high continuum optical depth derived from the FTS near 1 μm (~0.05 optical depths, see Section 3) is regarded as an undiagnosed issue (potentially due to the reasons postulated above) with the instrument sensitivity at high wavenumbers,



and henceforth we focus on the 1.6, 2.1 and 4 μm windows. This is additionally motivated by the lack of laboratory measurements to validate in the larger-wavenumber windows. However, we cannot rule out that the large observed optical depth is some unexplained physical effect (or indeed an unexpectedly-large water vapour continuum signal). Further clear-sky observations in this spectral region could affirm whether this is the case.

## 3.    Results

### 3.1: Best estimate from Langley measurements of 18 September 2008 and comparison with closure measurements

Figure 7 shows the best estimate (henceforth referred to as "CAVIAR-field") of our continuum from 21 observations on 18 September 2008 using the Langley method. Also shown are the MT_CKD 3.2 and 2.5 modelled continuum optical depth (self + foreign) for atmospheric conditions on this day. Since the uncertainties in our observations are large, there is agreement with

10 MT_CKD 3.2 and 2.5 within the $k = 2$ uncertainty limits in the centres of the 4, 2.1, 1.6 and 1.3 μm windows. Note that the MT_CKD continuum does not provide any uncertainties. The comparison between CAVIAR-field and MT_CKD will be discussed further in Section 3.2. Section 4 focuses on a comparison of this data to the available laboratory data. This Section demonstrates the consistency between the closure and Langley-derived data, which are quasi-independent methods of deriving the continuum (see Supplementary for more details). The Supplementary also includes a comparison of the 18 September best

estimate to data from other days from the field campaign, which were less suitable for analysis of the continuum due to measurement issues, increased aerosol extinction and lack of data availability.


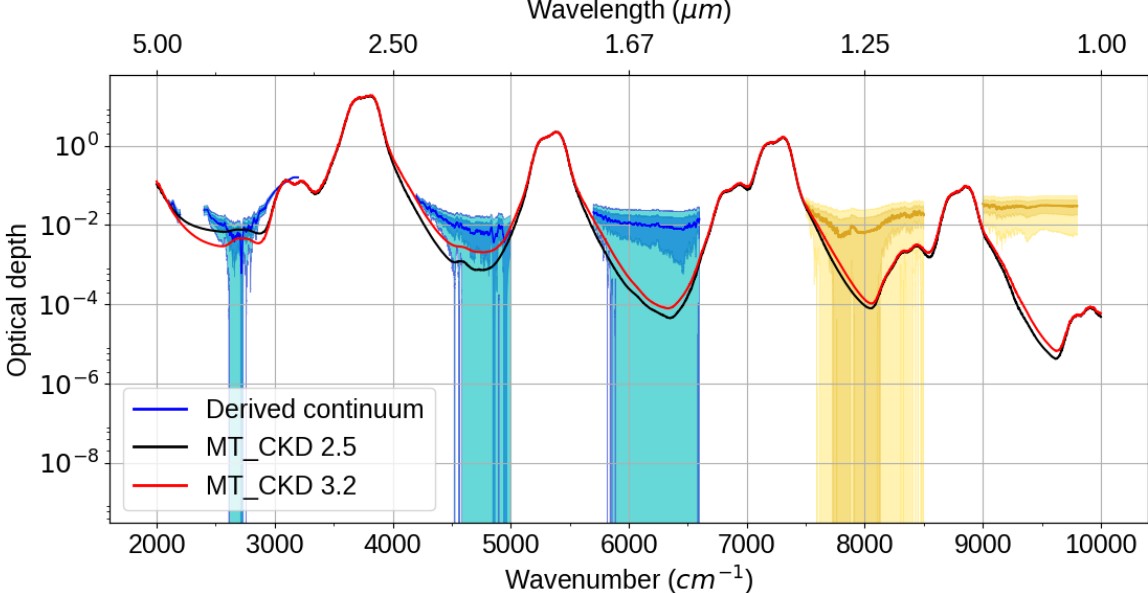

**Figure 7: Langley-derived CAVIAR-field continuum optical depth and optical depth for two versions of the MT_CKD water vapour continuum for 18 September 2008. The blue shaded regions indicate the $k = 1$ uncertainties, the cyan regions indicate the $k = 2$ uncertainties. The yellow shaded areas indicate spectral regions in which the CAVIAR-field derived continuum is potentially spurious and should be treated with caution (see Section 2).**

Figure 8 shows the comparison of the Langley-derived and closure-derived spectra from 18 September 2008. As with the Langley-derived spectrum, the closure-derived spectrum is a mean of 21 spectra from this day. The green and red lines overlap significantly on this Figure, indicating that there is excellent agreement between the two quasi-independent methods. This provides additional confidence in the accuracy of the Langley retrieval. The uncertainty in the closure-derived spectra is significantly larger, due to the use of an assumed SSI (from Elsey et al., (2017)) which itself has uncertainties.





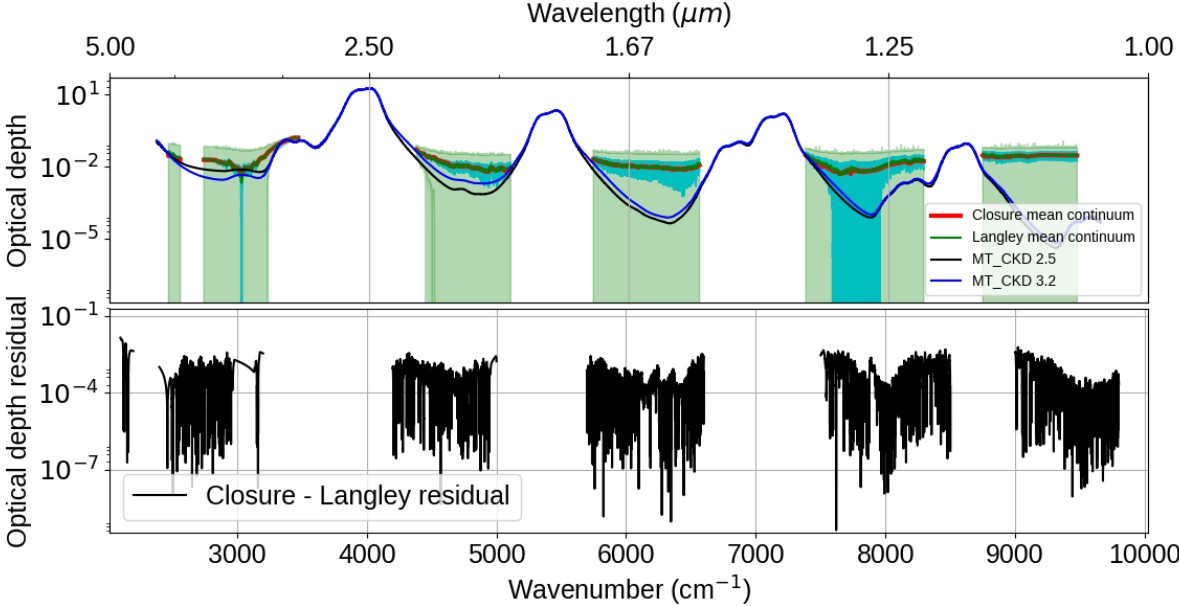

**Figure 8: Comparison between the Langley and closure method derivations of the continuum optical depth on 18 September 2008. Panel a) shows absolute values and b) the residual of the two. Teal shaded region is the $k = 1$ Langley uncertainty, green is the $k = 1$ closure uncertainty.**

One way of assessing any potential aerosol contamination is to look at the spectra at individual times, rather than the day-averaged continuum from the Langley method or the mean continuum as measured via the closure method. The closure-derived continua are calculated with aerosol extinction subtracted as observed by the Microtops at the time of each measurement. They are shown at different times across 18 September 2008 in Figure 9. Figure 9a shows the case with time-averaged aerosol as observed by the Microtops, and Figure 9b the case with time-varying aerosol. The uncertainties are not plotted for clarity, but

are large ($\pm$ 0.04), meaning that, despite the observed differences, the observations are consistent. Therefore, this change in aerosol over the day cannot be confirmed with any degree of significance; it is difficult to tell if observed differences in central values are real or a consequence of the uncertainties.

Assuming that the central values are well characterised, they show that the derived continuum (+ residual aerosol contribution)

increases by a factor of two across the day. It is clear from Figure 6b that the time variation in the aerosol extinction is not observed by the FTS. When using a time-averaged aerosol (Figure 9a), the different closure spectra are much more consistent. The agreement between the Langley and closure-derived continua in this case indicates that there are not significant issues with calibration of the instrument, unless such issues were strongly time-varying.





**Figure 9: Observed time-varying continuum optical depth derived from the closure method at different times throughout 18 September 2008. a) with time-averaged aerosol as observed by the Microtops and b) with time-varying aerosol. The uncertainties are not shown for visual clarity, but are on the order ~0.04 ($k = 1$).**

Given the level of uncertainty in these results, it is not certain if the differences between Figure 9a and 9b are significant. However, one possible source of difference that was considered was inaccuracy in the external mirror reflectivity correction (explained in more detail in Gardiner et al., 2012). However, as discussed in the Supplementary, a change in the reflectance will not lead to any change in the slope of a Langley fit, and therefore not impact the Langley-derived continuum in any way, provided the change is independent of angle. The Supplementary shows that this cannot account for the optical depth in the



higher-wavenumber windows without an undiagnosed change in the irradiance with angle, e.g. due to uncertainties in the phase correction as discussed in Section 2.

## 3.2: Comparison with MT_CKD

Figure 7 shows that, in the centre of the 4 μm window, the CAVIAR-field continuum optical depth appears to be in reasonable

agreement with the optical depth obtained using MT_CKD (and in better agreement with version 2.5 than version 3.2), but less so toward the edges of the window. This is further demonstrated in Figure 10, which shows the ratio of the CAVIAR-field continuum to two versions of MT_CKD. The agreement at the centre of the window is indicative of agreement with various FTS measurements in this region; this will be explored further in Section 4. At the higher-wavenumber edge however, there is no agreement between MT_CKD and CAVIAR-field within the $k = 2$ uncertainties; if our measurements are accurate (and in

agreement with other datasets), this indicates a strengthening to MT_CKD is required in that region.

In the 2.1 μm window, CAVIAR-field is inconsistent with MT_CKD within the $k = 1$ uncertainties in a significant portion of the window, and inconsistent within the $k = 2$ uncertainties in the lower-wavenumber part of the window. The ratio of CAVIAR-field to MT_CKD3.2 is ~5 in this region (Figure 10) but this is in significantly better agreement than would be the

case using the older MT_CKD2.5 values; this implies that either or both of the MT_CKD self and foreign continua need to be strengthened. Section 4 will further discuss the relative contribution of the self and foreign continua to this discrepancy.

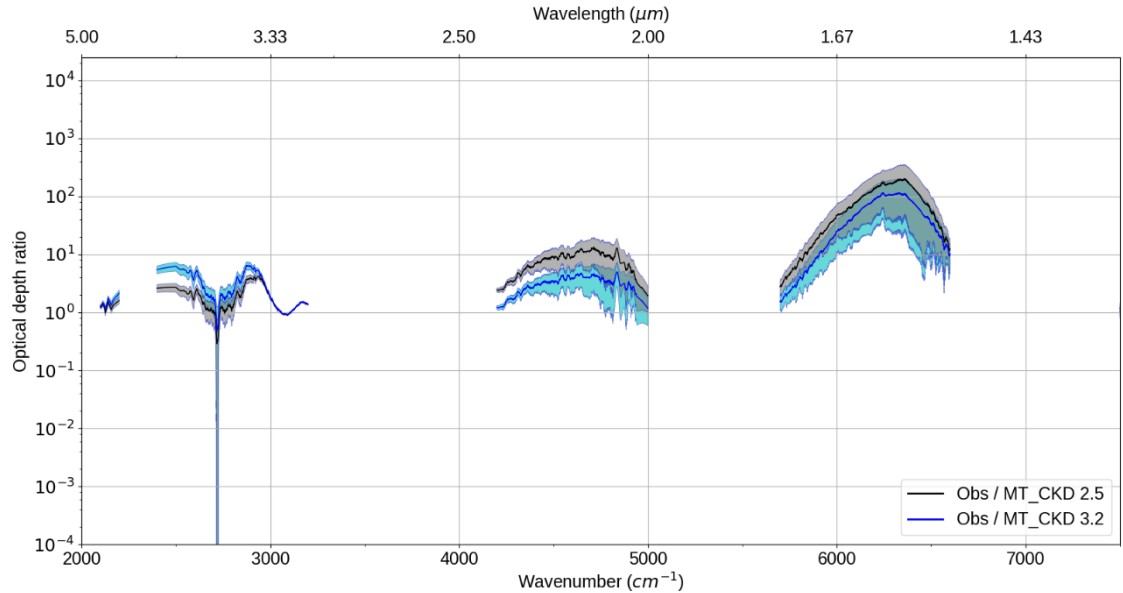

**Figure 10: Ratio of Langley-derived CAVIAR-field continuum optical depth divided by MT_CKD optical depth (for two different versions of MT_CKD) in the 2.1 and 1.6 μm atmospheric windows for 18 September 2008. Uncertainties (shaded regions) shown at**

**$k = 1$.**



The CAVIAR-field continuum is significantly stronger than either of the recent versions of MT_CKD within the 1.6 $\mu m$ window, disagreeing within the $k = 1$ uncertainties, but consistent within the $k = 2$ uncertainties. This is the region in which there is the most difference between available laboratory spectra. Section 4 will discuss this in more detail.

### 4. Comparison with laboratory observations

This section describes the relevant laboratory measurements in each of the 4, 2.1 and 1.6 μm atmospheric windows, and compares them to the continuum absorption from this work. The continuum absorption derived in Section 3 is difficult to directly compare to the laboratory measurements of the continuum absorption cross-section, since our derived continuum optical depth $\tau_{total}^{CAV}$ is the sum of the self + foreign continuum optical depths. However, it is possible to compare these data indirectly, via their ratio to MT_CKD3.2 using atmospheric conditions at the time of the measurements. Since:

(3) $\quad \tau_{tot}^{CAV} = \tau_{self}^{CAV} + \tau_{for}^{CAV}$

either the self or foreign continuum coefficient can be estimated by subtracting the optical depth contribution from the other. Consider the case in which $C_{self}^{CAV}$ (i.e. the CAVIAR-field self-continuum cross-section) is to be obtained. The foreign continuum optical depth $\tau_{for}^{CAV}$ is an unknown which must be estimated. This can be done by assuming either a) $\tau_{for}^{CAV} = \tau_{for}^{MT\_CKD}$ (the MT_CKD3.2 foreign continuum optical depth derived for the conditions of 18 September 2008), or b) by assuming

(4) $\quad \tau_{for}^{CAV} = \dfrac{C_{for}^{lab}}{C_{for}^{MT\_CKD}} \tau_{for}^{MT\_CKD}$

where $C_{for}^{lab}$ is the foreign continuum cross-section from laboratory observations and $C_{for}^{MT\_CKD}$ is the MT_CKD3.2 foreign continuum cross section. The ratio $\dfrac{C_{for}^{lab}}{C_{for}^{MT\_CKD}}$ effectively scales the MT_CKD optical depth to the laboratory observations. This is shown visually in Figure 11.

There is no constraint on the total optical depth $\tau_{tot}^{CAV}$ from the laboratory observations or MT_CKD. Taking $\dfrac{C_{for}^{lab}}{C_{for}^{MT\_CKD}}$ = the lab scaling factor $k_f$ (for the foreign continuum), with a corresponding scaling factor $k_s$ for the self-continuum:

(5) $\quad \tau_{tot}^{CAV} = a k_s \tau_{self}^{MT\_CKD} + b k_f \tau_{for}^{MT\_CKD}$

These parameters $a$ and $b$ determine the relative contribution of the self and foreign continua to the offset between the CAVIAR-field optical depth, and the optical depth from MT_CKD (with or without the laboratory scaling). However, as $a$ and $b$ are both unknowns, it is not possible to estimate one or the other without making some assumptions. In this analysis, we therefore assume that $b = 1$ when estimating the self-continuum, and that $a = 1$ when estimating the foreign-continuum, i.e. that the MT_CKD (with or without scaling to the laboratory observations) optical depth accurately represents the self or foreign component that is to be subtracted from the total to estimate the foreign or self-component respectively.





As will be demonstrated, for the conditions observed on 18 September 2008, both the self and foreign continua make significant contributions in the various windows to the total continuum. This is strongly dependent on whether MT_CKD or the laboratory data is used to estimate the self or foreign continua.

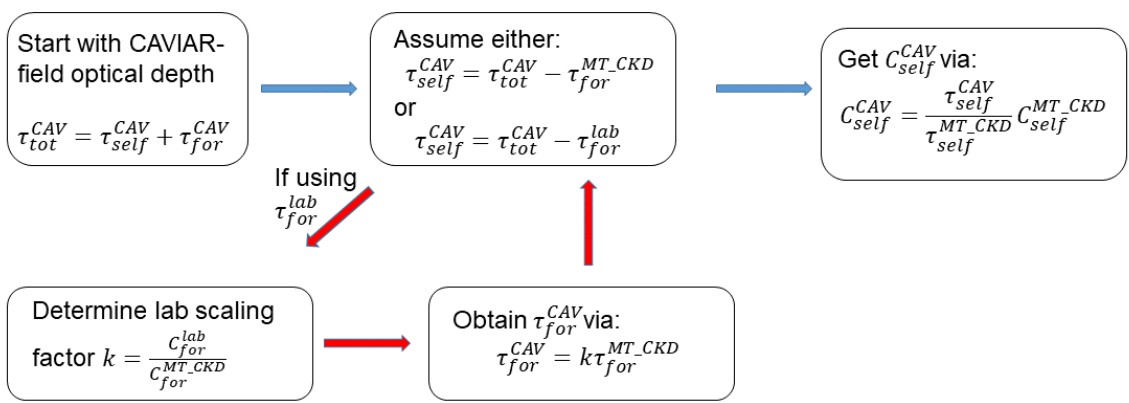

**Figure 11: Derivation process of the self-continuum absorption cross-section implied by the FTS optical depth. This process can be done for the foreign continuum, in which case all references to the self-continuum in the Figure apply to the foreign continuum and vice versa.**

This approach is reasonably robust when estimating the self-continuum, as this relies on the laboratory or MT_CKD foreign continuum, which is believed to be relatively independent of temperature (Ptashnik et al. 2012, Baranov 2011). Therefore, high-temperature laboratory foreign continuum measurements (with their lower uncertainties) can be used. In this case, we use the average of the 350 K, 372 K, 402 K and 431 K foreign continuum measurements of Ptashnik et al. (2012), henceforth the CAVIAR-lab foreign continuum, and assume that it is entirely independent of temperature. However, a lack of broadband room temperature measurements of the foreign continuum in the windows means that there may be some additional (and unquantifiable) uncertainty arising from this assumption.

When estimating the CAVIAR-field foreign continuum, due to the lack of laboratory measurements at atmospheric temperatures (i.e. below room temperature), one must assume a temperature dependence for the self-continuum. This is done by extrapolating the high-temperature laboratory data to atmospheric temperature either by a statistical fit when scaling to the laboratory data, or by relying on the MT_CKD temperature dependence when scaling to MT_CKD. This statistical fit assumes that the temperature dependence is proportional to $\exp(\frac{D_0}{T})$, where $D_0$ could be interpreted as relating to the dissociation energy of a water dimer (e.g. Ptashnik et al. (2011a)). For the axes used in Figures 13, 14 and 15 (1000/T vs. the logarithm of the continuum cross-section), this shows as a straight line. This temperature dependence is an assumption; this may break down at lower temperatures due to e.g. a change in regime from bound to quasi-bound dimers with increasing temperature (Ptashnik et al. (2011b, 2019)). We apply this temperature dependence to the self-continuum measurements of Ptashnik et al. (2011a), henceforth the CAVIAR-lab self-continuum. This dataset was chosen due to its wide spectral coverage and range of



temperatures, making it more suited to such an extrapolation, rather than using the room-temperature CRDS and OF-CEAS data (Section 4.1), where there are measurements at room temperature, but only at specific wavenumbers.

### 4.1: Self-continuum

When deriving the CAVIAR-field self-continuum this way, a representative temperature must be chosen to compare to the laboratory measurements, since the continuum observed by the FTS is the integrated continuum across the entire temperature and pressure range of the atmosphere. Figure 12 shows the fractional contribution to the total continuum optical depth from the surface upwards for selected wavenumbers, as calculated using MT_CKD3.2 and RFM for the conditions of 18 September 2008 at Camborne. This is calculated as the fractional contribution at each layer as observed by our radiosonde profiles to the total continuum absorption. This shows that more than 95% of the continuum optical depth is in the bottom 2 km of the atmosphere. This corresponds to a temperature range of ~275-290 K, which was taken to be representative of the temperature for which the CAVIAR-field self-continuum is representative.

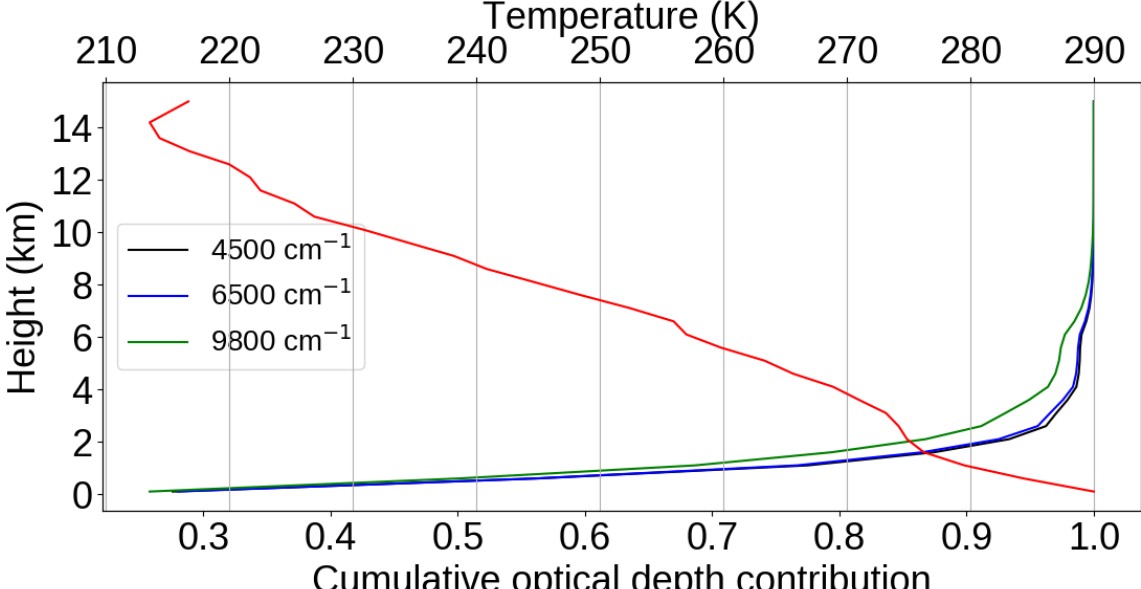

**Figure 12: Fractional contribution to the total continuum optical depth (using MT_CKD 3.2) with height from the surface up to 10 km at three wavenumbers. The temperature profile (top axis) derived from radiosonde on the 18 September 2008 is shown in red.**

Shine et al (2016) presents a review of the laboratory data up to 2016 in significant detail. The following paragraphs introduce the main data available across multiple spectral windows used to compare the CAVIAR-field self-continuum. Other datasets will be introduced as required for each specific window.



Ptashnik et al., (2011a) (CAVIAR-lab self-continuum) presented laboratory observations of the self-continuum taken by a Fourier transform spectrometer (FTS) from 472-293 K between 2500-10000 cm$^{-1}$; because uncertainties become too large for the low temperature measurements (because of the lower vapour pressures that are necessary at low temperatures), at wavenumbers greater than 5600 cm$^{-1}$ measurements are restricted to 374 K and above, with the exception of a few 350 K

measurements at the edges of the windows where the continuum is stronger. However, at all wavenumbers, uncertainties are larger at the lower temperatures. Further sets of FTS measurements (Ptashnik et al., 2013, 2015) were taken at the Institute for Atmospheric Optics in Tomsk, Russia. However, recent and ongoing study has indicated that these results may be spurious, due to reflectivity issues arising from adsorption onto the gold mirrors used in their multipass cell (Ptashnik et al. (2019b), https://symp.iao.ru/files/symp/hrms/19/en/abstr_11485.pdf). For this reason, these results have not been included in our

analysis.

Various sets of observations have been made by the Spectroscopy Group at LiPhy (Laboratoire Interdisciplinaire de Physique) at the Université Grenoble-Alpes (henceforth collectively "Grenoble"). Mondelain et al., (2013, 2014) presented observations of the near-IR self-continuum in the 1.6 and 2.1 μm windows at room temperature and above using a cavity ring-down

spectrometer (CRDS). Newer measurements (Lechevallier et al., 2018; Richard et al., 2017) by this group in the 2.1 μm window were presented, which generally agree with the Mondelain et al. observations. Vasilchenko et al., (2019) present updated CRDS measurements at a range of wavenumbers at room temperature in the 2.1 and 1.6 μm windows. Ventrillard et al., (2015) present observations in the 2.1 μm window (4302 and 4732 cm$^{-1}$) using an optical feedback cavity enhanced absorption spectroscopy (OF-CEAS) technique at w (293-323 K), while Richard et al. (2017) use the same technique in the 4

μm window (2491 cm$^{-1}$).

### 4.1.1: 4 μm window

Figure 13 presents various estimates of the self-continuum from the laboratory in the centre of the 4 μm window (2491 cm$^{-1}$). The CAVIAR-lab measurements agree reasonably well at ~350 K with the laboratory FTS data of Baranov and Lafferty (2011), taken at several temperatures across the 4 μm window. However, there is poor agreement between these FTS data and both

the measurements of Richard et al. (2017), and grating spectrometer measurements of Burch and Alt (1984). The Richard et al. (2017) and Burch and Alt (1984) agree reasonably well and imply a weaker temperature dependence than CAVIAR-lab and Baranov and Lafferty (2011). However, extrapolating only through the high-temperature CAVIAR-lab data (i.e. excluding the point at 293 K) using an assumed $\exp(\frac{D_0}{T})$ temperature dependence yields excellent agreement with the Burch and Alt (1984) and Richard et al. (2017) data. These higher-temperature CAVIAR-lab measurements have smaller uncertainties than

the low-temperature CAVIAR-lab measurements, and appear to lie on a straight line, while the lower-temperature CAVIAR-lab data point does not lie on this line. There are therefore two possible experimentally-implied temperature dependences, a lower one implied by the high-temperature CAVIAR-lab, Richard et al. (2017) and Burch and Alt (1984) measurements, and



a stronger dependence implied by the less-certain lower-temperature CAVIAR-lab, and Baranov and Lafferty (2011) measurements. This reflects the importance of making observations at lower temperature with well-constrained uncertainty budgets.

Figure 13 also shows the estimated CAVIAR-field self-continuum, obtained using the algorithm presented in Figure 11. This is estimated by subtracting the MT_CKD foreign continuum from the observed total continuum (orange point), or by subtracting the CAVIAR-lab foreign continuum (green point). When assuming a foreign continuum from MT_CKD, the CAVIAR-field estimated data agrees better with the low-temperature FTS data, whereas assuming the CAVIAR-lab foreign continuum leads to somewhat better agreement with the high-temperature extrapolated CAVIAR-lab, Burch and Alt (1984)
and Richard et al., (2017) data, although it is about a factor of 2 below the extrapolated line. This may indicate that the CAVIAR-lab foreign continuum is too large, or that the CAVIAR-field derived continuum is too weak. Interestingly, the MT_CKD3.2 self-continuum is inconsistent at the $k = 1$ level with both of the CAVIAR-field data points at these temperatures, being a factor of 4 stronger than the estimate using the CAVIAR-lab foreign continuum and a factor of ~5 weaker than the estimate using the MT_CKD foreign continuum. This shows the importance of the assumption of a given foreign continuum
to this analysis. Given the high precision of the Richard et al. measurements, the anomalous deviation from the assumed temperature dependence from the CAVIAR-lab data point at 297K, and the CAVIAR-field estimate being closer to this temperature dependence when using the CAVIAR-lab foreign continuum, we believe that this provides evidence for the temperature dependence being lower than implied by Baranov and Lafferty (2012). We note however that there is no agreement between the $k = 1$ CAVIAR-field uncertainty limits and the straight line of the fit to CAVIAR-lab, or Richard et al. (2017).



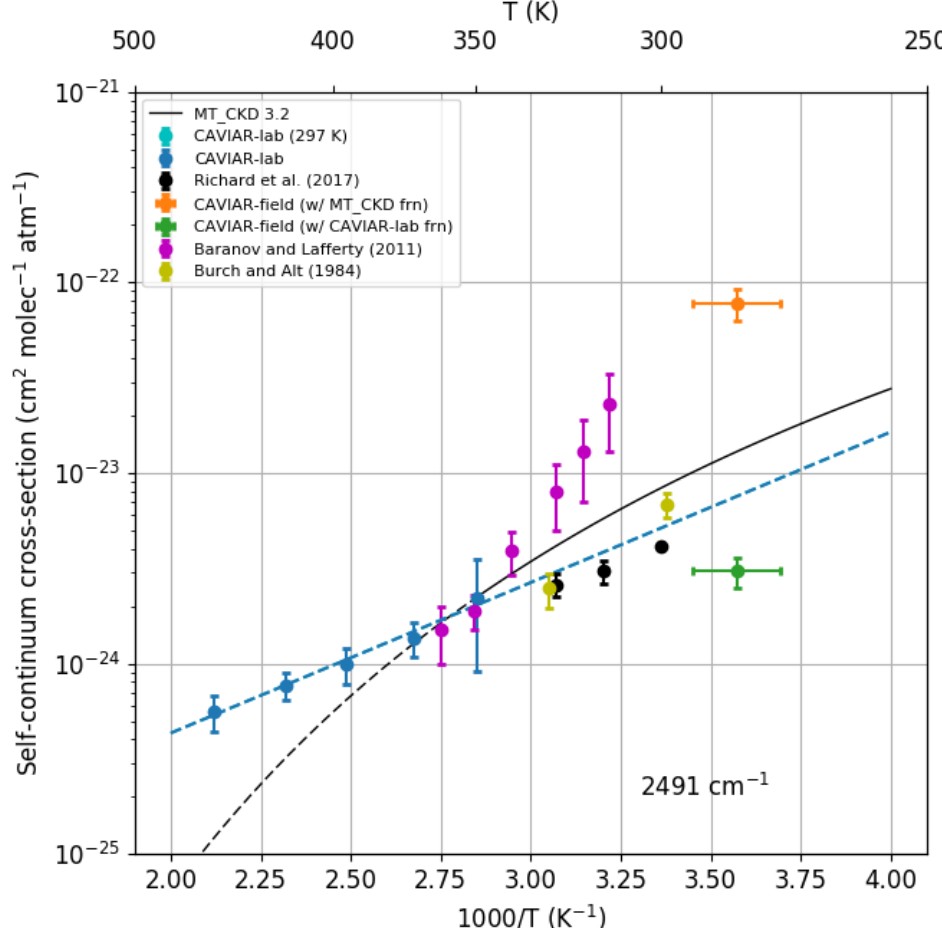

**Figure 13: Self-continuum absorption cross-section against temperature for various datasets at 2491 cm⁻¹. The error bars indicate the k = 1 uncertainties. Marker size is greater than the stated uncertainty where error bars are not visible. The dashed black line of MT_CKD above 350 K indicates the region outside of the expected applicability of MT_CKD. The dashed blue indicates extrapolation of the high-temperature CAVIAR-lab temperature dependence, while the cyan dashed line shows the extrapolation through all of the CAVIAR-lab data points (including the low-temperature cyan data point).**

### 4.1.2: 2.1 μm window

At 2.1 μm (Figure 14), there is generally good agreement between the various sets of laboratory data, particularly when extrapolating the high-temperature CAVIAR-lab data to room temperature (blue dashed line), rather than using the lower temperature data (cyan point and dashed line) which has larger uncertainties. There is also good agreement between these laboratory data and MT_CKD, within the temperature range in which MT_CKD is expected to be valid (solid line). Figure 14 also shows the estimated CAVIAR-field at 3 wavenumbers at the edge (4255 and 4302 cm⁻¹) and centre (4723 cm⁻¹) of the 2.1 μm window. This is derived using both the MT_CKD3.2 foreign continuum (orange data point) and the CAVIAR-lab foreign





continuum (green data point). The error bars show the $k = 1$ uncertainty limits. The error bars on the Mondelain et al. (2015) and Ventrillard et al. (2015) measurements are smaller than the marker size.

At the edge of the window (Figure 14 a, b), the CAVIAR-field estimated continuum (assuming the MT_CKD foreign

continuum) does not overlap with the available laboratory data within the $k = 1$ uncertainties. This suggests that, if the available data are robust and the assumed temperature dependence of the self-continuum is correct, the MT_CKD foreign continuum requires some strengthening at the edge of this window. However, using the CAVIAR-lab foreign continuum at the low-wavenumber edge of the window results in a negative implied self-continuum, indicating that either the CAVIAR-lab foreign continuum is too strong at the window edge, that there is a temperature dependence of the foreign-continuum that is neglected

here, or that the observed optical depth from this work is poorly characterised at the edge of the window. In the centre of the 2.1 µm window (Figure 14c) the CAVIAR-field estimated continuum shows reasonable agreement with the observed laboratory data (extrapolated to 280 K) when using the CAVIAR-lab foreign continuum, with overlap between the $k = 1$ uncertainties. This is not the case when using the MT_CKD foreign continuum, providing further evidence that this requires strengthening, particularly at the centre of the window. This is consistent with laboratory analyses of the foreign continuum

(Section 4.2).

It is important to note that there is good consistency between the 297K CAVIAR-lab data point and the CAVIAR-field estimated self-continuum when using the MT_CKD foreign continuum. This would suggest that if the self-continuum is as large as implied by this lower-temperature data point, the foreign continuum would be robust in MT_CKD 3.2. However,

given the agreement between the high-temperature CAVIAR-lab data (which have lower uncertainties), and the laboratory foreign-continuum data available in the 2.1 µm region (Section 4.2), we believe that the likelihood is that the foreign continuum requires strengthening rather than the self-continuum.

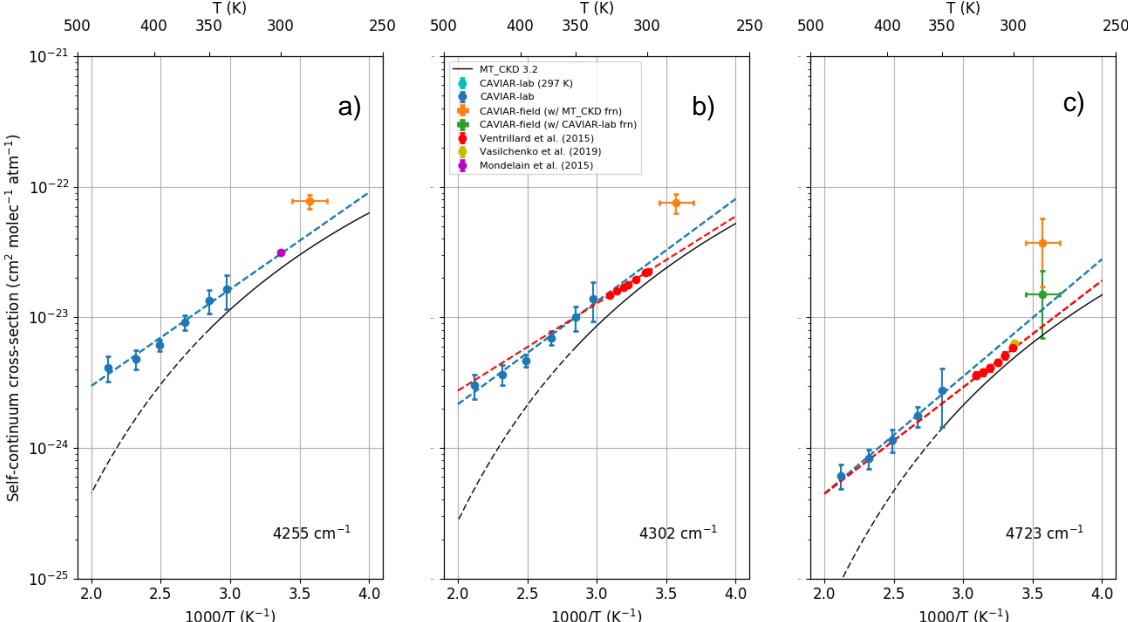

**Figure 14: Self-continuum absorption cross-section against temperature for various datasets at (a) 4255, (b) 4302 and (c) 4723 cm⁻¹. The error bars indicate the $k = 1$ uncertainties. Marker size is greater than the stated uncertainty where error bars not visible. The dashed line of MT_CKD above 350 K indicates the region outside of the expected applicability of MT_CKD. The dashed blue, cyan and red lines indicate extrapolations of the CAVIAR-lab (with and without the low temperature data) and Grenoble temperature dependence respectively, assuming an exponential temperature dependence. The green (CAVIAR-field with CAVIAR-lab foreign continuum) are missing from frames a) and b) as the inferred self-continuum is negative.**

### 4.1.3: 1.6 µm window

Figure 15 shows the observed absorption cross-section as a function of temperature for three wavenumbers in the 1.6 µm window (6050, 6177 and 6383 cm⁻¹) in panels a), b) and c) respectively. These wavenumbers were selected since these are the wavenumbers in which the Grenoble CRDS data is available. The agreement in this window is generally lacking between different laboratory datasets of the self-continuum. At room temperature, the extrapolated CAVIAR-lab data imply significantly stronger absorption than the Grenoble data (from Mondelain et al., (2014) and Vasilchenko et al., (2019)). The data indicate significantly different temperature dependences between CAVIAR-lab and Grenoble; the latter shows a significantly weaker temperature dependence across the window, and a weaker dependence relative to CAVIAR-lab than the Grenoble measurements in the 2.1 µm window (Figure 14). Both the Grenoble and CAVIAR-lab temperature dependences are markedly different from the MT_CKD3.2 temperature dependence. These discrepancies are discussed in Shine et al., (2016). In addition to the data discussed at the beginning of Section 4, we present an additional comparison at 6177 cm⁻¹ (Figure 15b) with the continuum derived in Kapitanov et al., (2018) using a photo-acoustic method. An additional issue arises when comparing the Mondelain et al. and Vasilchenko et al. data; while the observed absorption cross-sections are similar, there is no agreement within their stated uncertainties. This can be attributed to the differences in the fits used to obtain these cross-





sections; both used a quadratic fit of vapour pressure versus absorption to obtain their cross-sections, but Mondelain et al. use an additional linear term to account for supposed adsorption on the mirrors, whereas Vasilchenko et al. did not need to use this additional term The Vasilchenko et al., (2019) data, being more recent, is regarded as the more reliable, but is only available at one temperature.

The choice of foreign continuum has less of an effect in this window since the absolute difference between MT_CKD and CAVIAR-lab foreign continuum is too small to significantly affect the large observed optical depth. However, since the associated uncertainties are large (the $k = 2$ uncertainties intersect with zero), they are not entirely inconsistent with any of the observed data. The Grenoble measurements imply an extremely weak temperature dependence which is inconsistent with that of either CAVIAR-lab or MT_CKD3.2, and less consistent with the estimated CAVIAR-field data than CAVIAR-lab. While these results indicate a significantly stronger continuum than that implied by the available laboratory data, the uncertainties are too large to form firm conclusions. In addition, the CAVIAR-field results do not reconcile the apparent large discrepancy between the extrapolated CAVIAR-lab continuum and the Grenoble measurements; this contrasts markedly with the situation in the centre of the 2.1 window (Fig. 14c), where there is consistency between these datasets and reasonable consistency with CAVIAR-field, when the CAVIAR-lab foreign continuum is used.

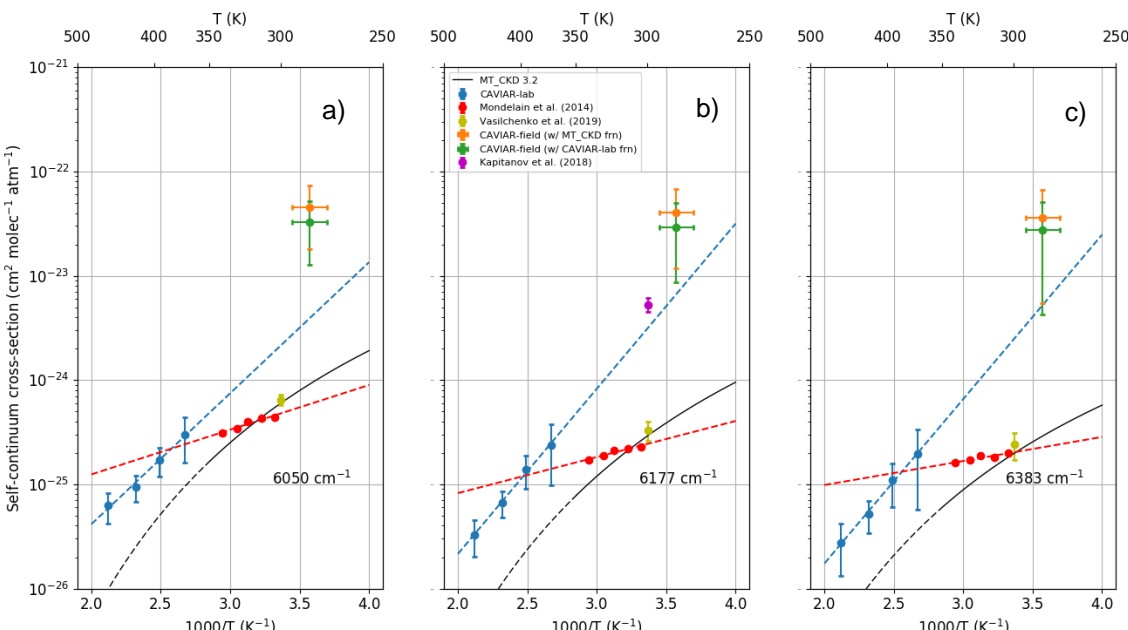

**Figure 15: Self-continuum absorption cross-section against temperature for various datasets at (a) 6050, (b) 6177 and (c) 6383 cm⁻¹. The error bars indicate the $k = 1$ uncertainties. Marker size is greater than the stated uncertainty where error bars not visible. The dashed black line indicates the region outside of the expected applicability of MT_CKD. The dashed blue and red lines indicate extrapolations of the CAVIAR-lab and Grenoble temperature dependence respectively.**





### 4.1.4: Synthesis

Figure 16 shows the spectrally-resolved self-continuum from CAVIAR-field (using the two foreign continua), alongside other sets of observations. The CRDS measurements are shown at their original temperature, since for many wavenumbers observations are only available at one temperature. Assuming the MT_CKD temperature dependence holds, these can be scaled

5    by a factor of ~1.35 to be brought to 280 K. This shows that the estimated continuum does not vary particularly strongly spectrally in the centres of the windows. However, there are clearly some issues in the 4 µm window, where at lower wavenumbers the derived continuum is significantly stronger than MT_CKD3.2 and the extrapolated CAVIAR-lab self-continuum, and in the low-wavenumber edge of the 2.1 µm window, where when estimating the self-continuum using the CAVIAR-lab foreign continuum there is a significant decrease in the self-continuum. As discussed previously, this is likely

10    due to either the CAVIAR-lab foreign continuum being too strong at this low-wavenumber edge, or some issue with the FTS field observations used in this work. In the 1.6 µm window, a significant strengthening of the foreign continuum of over a factor of 100 would be required to bring the CAVIAR-lab and CAVIAR-field self-continua into agreement, which is inconsistent with the CAVIAR-lab foreign continuum in this window.





**Figure 16: Self-continuum from CAVIAR-field as estimated using (a) the MT_CKD foreign-continuum and (b) the CAVIAR-lab self-continuum, alongside MT_CKD3.2 and selected laboratory measurements. The shaded regions indicate the $k = 1$ confidence limits. The CAVIAR-lab self-continuum is derived from extrapolating the high temperature (>350 K) data to 280 K. The uncertainties are obtained by scaling the uncertainties at higher temperatures with this same extrapolation.**



### 4.2: Foreign continuum

#### 4.2.1: Current observations

Ptashnik et al. (2012) (CAVIAR-lab) presented foreign continuum coefficients in the 4, 2.1, 1.6 and 1.2 μm windows using FTS; these remain the only laboratory dataset with a large wavenumber coverage. These observations are made using a cell

filled with an $H_2O$-air mixture, and then subtracting the self-continuum contribution as measured by Ptashnik et al. (2011a). At all temperatures, their foreign continuum is consistently stronger than all versions of MT_CKD in the central parts of the 4-1.6 μm windows (between 10-100 times stronger than MT_CKD2.5), although there is better agreement at the edges of these windows.

Baranov and Lafferty (2012) report foreign continuum values in the 4 μm window using an FTS technique, assuming the self-continuum as measured by Baranov and Lafferty (2011). These agree very well with the measurements of Baranov (2011), which were taken at four temperatures (326, 339, 352 and 363 K). In a similar way to the CAVIAR-lab foreign continuum, they observe a factor of 100 stronger foreign continuum than MT_CKD2.5 in the centre of the 4 μm window, and are in reasonable agreement with Ptashnik et al. (2012). They also exhibit no significant temperature dependence, in agreement with

Ptashnik et al. (2012).

Mondelain et al., (2015) presented a foreign continuum measurement at one wavenumber (4250 cm[-1]) at 298 K using the CRDS technique. Their reported values were a factor of ~4.5 stronger than MT_CKD in this region, and a factor of ~2 weaker than CAVIAR-lab. Vasilchenko et al., (2019) present foreign continuum data at 4435, 4522, 4720 and 4999 cm[-1] using CRDS.

Their data does not agree within the $k = 1$ uncertainties of the CAVIAR-lab FTS measurements (aside from at 4720 cm[-1]) and are systematically lower by a factor of 2-4. They do however agree within the $k = 2$ uncertainties. The CRDS foreign continuum was measured at room temperature; assuming that both the FTS and CRDS measurements are robust, this would indicate a small positive temperature dependence for the foreign continuum.  The Vasilchenko et al. data is systematically a factor of 5 stronger than the MT_CKD foreign continuum; both the FTS and CRDS data indicate that MT_CKD therefore requires some

strengthening, but by differing amounts.

The only existing dataset of pure foreign continuum measurements in the 1.6 μm window is the CAVIAR-lab data. We therefore focus our comparison solely on MT_CKD and CAVIAR-lab in this region.

In addition to the laboratory measurements, Reichert and Sussmann (2016) presented measurements of the water vapour continuum in the atmosphere between 2500-7600 cm[-1] (see Section 1.2 for more details). Given the high altitude and low water vapour path of their measurements, Reichert and Sussmann indicate that the foreign continuum is by far the dominant contributor to the continuum in the majority of their measured spectral regions. This is particularly likely to be the case in the

atmospheric windows, where the foreign continuum contribution is larger (e.g. Ptashnik et al. 2012). Reichert and Sussmann present data in the 4, 2.1 and 1.6 µm windows; however due to the low atmospheric absorption seen in their experiment the results are negative for a significant portion of the spectrum. However, their uncertainty limits provide an upper bound on the strength of the foreign continuum.

**4.2.2: CAVIAR-field foreign continuum**

The foreign continuum can be inferred from the CAVIAR-field measurements using high temperature observations of the self-continuum extrapolated down to room temperature. This allows for comparison with the laboratory foreign continuum data, and with Reichert and Sussmann, (2016). Figure 17 shows the CAVIAR-field foreign continuum for two different cases; assuming a) the MT_CKD self-continuum, and b) the high-temperature CAVIAR-lab self-continuum extrapolated to 280 K.

In this case, only the data points above T = 297 K have been included in the extrapolation, to better reflect the agreement (when extrapolated) with the available Grenoble measurements in these windows, which have lower uncertainties at low-temperature.

We focus the discussion here on the 2.1 and 4 µm windows, since these are the regions in which the most laboratory data are

available. It is important to emphasise here that the foreign continuum cannot be derived from laboratory measurements without prior knowledge of the self-continuum, and that therefore the foreign continuum values shown are sensitive to the assumptions made about the self-continuum.

In the centre of the 4 µm window, Fig. 17 shows that the foreign continuum is significantly stronger (~20x) than MT_CKD

regardless of the assumption made about the self-continuum, and agrees well with the CAVIAR-lab and Baranov (2011) foreign continua, which are plotted here at 326 K. It is also consistent with Reichert and Sussmann (2016) within the $k = 1$ uncertainty limits. The weight of available data appears to indicate that a significant strengthening of the MT_CKD foreign continuum is required in the centre of this window. Given that Baranov and Lafferty (2012) retrieval of the foreign continuum uses the Baranov and Lafferty (2011) self-continuum, which may be an overestimate if the Grenoble measurements are correct

(Figure 13), this strengthening could even be larger than indicated.

Figure 17 shows that in the centre of the 2.1 µm window there is excellent agreement between the CAVIAR-lab and CAVIAR-field foreign continua whether using either the MT_CKD (panel a) or CAVIAR-lab self-continuum (panel b). This provides evidence that, assuming our knowledge of the self-continuum is robust, the foreign continuum is better characterised by

CAVIAR-lab than MT_CKD. As indicated in Reichert and Sussmann (2016), their values can only represent an upper limit on the continuum in the windows. Nevertheless, these results agree with ours within the $k = 1$ uncertainties, indicating that the two are consistent. However, at the low-wavenumber edge of the window (~4200 cm⁻¹), our results show a somewhat weaker (factor of ~2) foreign continuum than CAVIAR-lab. This is consistent with the inference made in Section 4.1.2 when





estimating the self-continuum. Our results are consistent with the uncertainty limits of Reichert and Sussmann (2016) at these wavenumbers. There is good agreement between the various laboratory self-continuum data in this window (when extrapolated to room temperature), which gives some confidence in the analysis presented in Figure 14. Our reported uncertainties are also smaller in this region, and any unattributed aerosol effect would be smaller in this window than at 1.6 μm. These results

5    indicate that the foreign continuum is stronger than the MT_CKD foreign continuum by about a factor of 5 in the centre of the window, in agreement with Ptashnik et al., (2012) and Vasilchenko et al. (2019).







**Figure 17: Langley-estimated foreign continuum across the 2000-7000 cm⁻¹ region alongside CAVIAR-lab, MT_CKD 3.2 and Reichert and Sussmann (2016) data. The shaded regions and error bars indicate the $k = 1$ uncertainties. Panel a) shows the CAVIAR-field foreign continuum assuming the MT_CKD3.2 self-continuum and panel b) shows the CAVIAR-field foreign continuum assuming the CAVIAR-lab self-continuum, which is derived from extrapolating the high temperature (>350 K) data to 280 K.**

The situation in the 1.6 μm window (Fig. 17) is less clear. The uncertainties in our measurements are greater, and there is less

consistency in this window between this work and the laboratory data. The agreement improves when the stronger CAVIAR-

lab self-continuum is used (Fig 17b). This could indicate that there is an issue with our measurements in this window (such as

aerosol contamination or a systematic calibration uncertainty), or that the foreign continuum is significantly stronger than





predicted by CAVIAR-lab. Despite the large observed values, CAVIAR-field, CAVIAR-lab and Reichert and Sussmann (2016) are all consistent within their $k = 2$ uncertainty limits. These results indicate a larger absorption than observed in the CAVIAR-lab data. Such a large absorption could explain the results of Oyafuso et al., (2017), who reported that "unrealistically large multiplicative factors (~8x for the 2.06 µm band and ~150x for the 1.6 µm band) for the water vapor continuum were required". This work strongly suggests that a strengthening of the foreign continuum by a factor of 10x is necessary to MT_CKD3.2 at 2.1 µm (consistent with laboratory observations), and absorption a factor of ~100x stronger than MT_CKD3.2 in the 1.6 µm window (which is less consistent with laboratory observations). There appears to be an urgent need for an independent set of foreign continuum measurements in the 1.6 µm window to resolve this discrepancy.

### 4.3: Relative contributions of the self and foreign continuum

An additional issue of importance is the relative contribution to the total continuum absorption of the self and foreign continua, particularly for atmospheric scientists, since the relative contribution of each is strongly dependent on the atmospheric conditions at the time of measurement. Figure 18 shows the percentage of the optical depth originating from the self and foreign-continua for conditions of 18 September 2008 from MT_CKD3.2 (the optical depth calculated using MT_CKD3.2 is shown in Figure 7). In these conditions (with an integrated water vapour column of about 16 kg m$^{-2}$, see Section 2), the self-continuum dominates in the centres of the windows (~95% in the centre of the 4 µm window, 90% in the centre of the 2.1 µm window and ~80% in the centre of the 1.6 µm window), while the foreign continuum dominates in the bands.

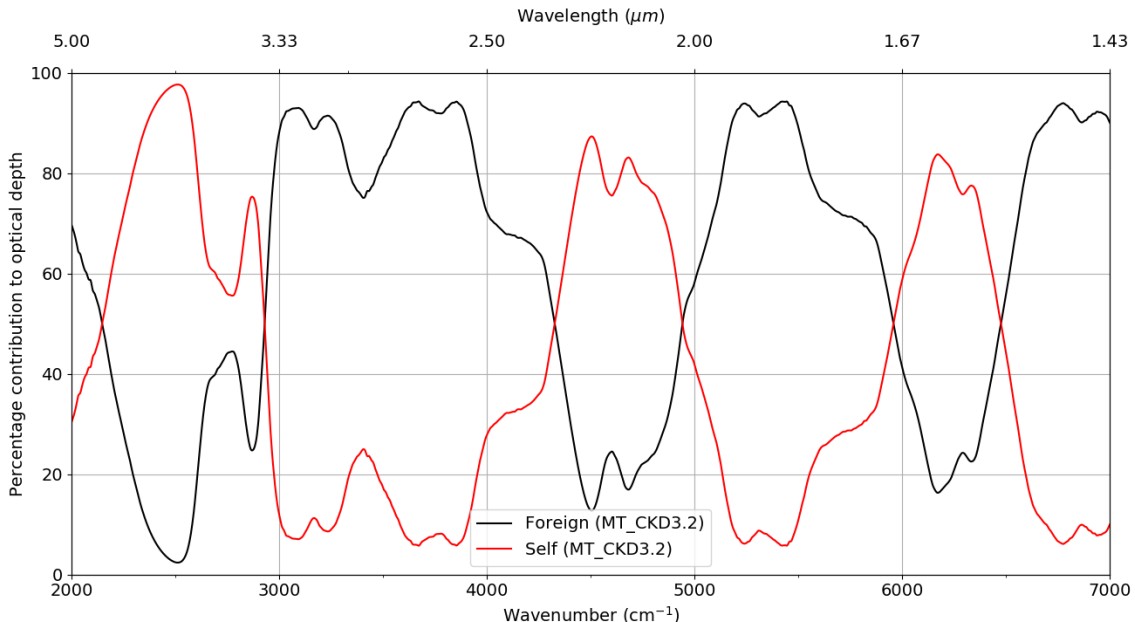

**Figure 18: Proportion of the 18 September 2008 Camborne optical depth attributable to the self and foreign continua as calculated using the MT_CKD3.2 model. The total optical depth is shown by the red line in Figure 7.**





Figure 19 shows the percentage contribution of the CAVIAR-field self-continuum (panel a) and CAVIAR-field foreign continuum (panel b). Each of these panels shows the proportion of the total 18 September 2008 continuum optical depth attributable to the self or foreign optical depth by assuming the contribution from the other component via either CAVIAR-lab or MT_CKD. The relative contribution in this case is as given by Eq. (5). Unlike with MT_CKD, which is well-constrained (and therefore the total contribution of the self and foreign continua sum up to the total, as in Figure 18), the CAVIAR-field estimated continuum is not, in the case where $a \neq b \neq 1$, as we do not have enough information to derive independent values of the two components of the CAVIAR-field continuum. Therefore, it should not be expected that $\tau_{for}^{CAV} + \tau_{self}^{CAV} = \tau_{tot}^{CAV}$. Figure 19 should be interpreted as the values implied by CAVIAR-field when assuming that the self or foreign contribution is well-characterised by either CAVIAR-lab or MT_CKD.

The self-continuum (panel a) contribution is large ($> 95\%$) when using the MT_CKD foreign continuum across all of the windows of interest, similar to the case shown in Figure 18. However, when using the CAVIAR-lab foreign continuum, this contribution decreases by an amount depending on the window of interest. In the 4 µm window, the contribution varies from ~50% to almost 0% in the centre of the window. Similarly, in the 2.1 µm window, the self-continuum drops from ~95% to ~40% contribution when using the stronger CAVIAR-lab foreign continuum. At 1.6 µm, almost all of the absorption when assuming the MT_CKD foreign continuum is implied to come from the self-continuum. This is because the MT_CKD3.2 foreign continuum is extremely weak in this region, and the total CAVIAR-field optical depth is much larger than the MT_CKD3.2 optical depth (see Fig. 7). Using the CAVIAR-lab foreign continuum decreases the contribution of the self-continuum to ~80% in the centre of this window, comparable with the fraction implied when just using MT_CKD (Figure 18).

Because of the lack of constraint on the CAVIAR-field optical depth, the implied foreign-continuum contribution when assuming the MT_CKD or CAVIAR-lab self-continua (Fig. 17b) is also high (over 60% across all 3 windows). Unlike the self-continuum case, there is reasonable consistency between the implied values using either CAVIAR-lab or MT_CKD. At 4 µm, using the CAVIAR-lab self-continuum increases the foreign contribution from ~60-80% to ~75-90% in the centre of the window, since the CAVIAR-lab self-continuum is smaller at room temperature than that of MT_CKD in this region (see Fig. 13). At 2.1 µm, the contribution drops from ~80% to ~70% when using the MT_CKD and CAVIAR-lab self-continua respectively. At 1.6 µm, the CAVIAR-field foreign is almost 100% in the centre of the window when using MT_CKD self-continuum, but drops to ~80-90% when using the CAVIAR-lab self-continuum.

The lack of consistency between the CAVIAR-field estimated self and foreign continuum is an indication of the lack of constraint on $a$ and $b$, meaning that there are potentially issues with CAVIAR-field, and/or with the laboratory measurements and/or MT_CKD.







**Figure 19: Proportion of the total continuum optical depth for 18 September 2008 from the CAVIAR-field self-continuum assuming the CAVIAR-lab and MT_CKD3.2 foreign continua (panel a), and the CAVIAR-field foreign continuum assuming the CAVIAR-lab and MT_CKD3.2 self-continua (panel b). As explained in the text, these is insufficient data to constrain the field observations such that the percentage contributions of the self and foreign sum to 100%, given the methodology to derive the self and foreign components of the field continuum.**



## 5. Future steps

Given the uncertainties present in this analysis, and the need to measure in a wider range of conditions to more accurately separate the foreign and self-continua, more measurements are required to sufficiently constrain the continuum absorption in atmospheric conditions. This section details how a future field campaign might reduce the uncertainty in the derived continuum

when performing an analysis such as the one presented in this work.

The main contributor to the uncertainty was the lack of well-characterised aerosol extinction. This is the most significant factor in the uncertainty budget, and there were significant problems in characterising the variation over time. This may have been due to operational problems with the Microtops sunphotometer used to measure aerosol optical depth. While the stated

uncertainty is reasonably small (e.g. Ichoku et al., (2002) estimate an optical depth uncertainty of ~ ± 0.02 in the lower wavenumber channels and ± 0.01 or less in the higher wavenumber channels) there was a clearly observed time variation in the $\tau_{aerosol}$ that was not present in the FTS measurements. Additionally, the observations of $\tau_{aerosol}$ were taken in channels in the visible and near-infrared parts of the spectrum and extrapolated out into the near-infrared. This means that, while the aerosol optical depth decreases with decreasing wavenumber (e.g. Figure 5), there is a higher fractional uncertainty since there

is a need to extrapolate further.

Ideally, any future campaign would use a more robust method of measuring aerosol extinction, such as taking place close to an AERONET site (e.g. Giles et al., 2019). A future campaign should minimise the aerosol contamination by taking place at higher altitude. High altitude observations would take place in the tropics, to ensure there is enough water vapour for the

continuum signal to be detectable in the windows. Additionally, satellite products could be used to measure $\tau_{aerosol}$; these have improved significantly in the decade since the Camborne observations were taken and could be used in conjunction with AERONET and in-situ measurements to constrain aerosol. Some caution should be warranted however, since satellites use atmospheric windows to obtain aerosol which also contain the ill-constrained continuum absorption.

Additionally, aircraft could be used to constrain the aerosol profile, aiding in e.g. calculations using a Mie scattering code. Measuring the aerosol profile in ambient conditions would be a significant step towards a more robust representation of $\tau_{aerosol}$, e.g. from a research aircraft. Aircraft could also be used to provide measurements of other variables, such as temperature, or even for fully radiometric measurements. Green et al., (2012) and Newman et al., (2012) measured the mid and far-infrared continuum via aircraft measurements during the CAVIAR project using the FAAM aircraft. This method

works well for measuring the comparatively strong mid and far-infrared continuum, but could potentially be used to measure the in-band continuum in the near-IR. However, this method relies on accurate calibration either to a blackbody source or to a prescribed SSI to retrieve the continuum via the closure method, since it is difficult to perform a Langley analysis using an aircraft.

Future campaigns could use the calibration method described in Reichert et al., (2016) to calibrate a spectrometer to the top of atmosphere solar irradiance, rather than using a comprehensive radiometric calibration such as that used in this work, once it is known to higher accuracy. This would reduce the costs of such a campaign, and potentially allow for observations in a wider range of conditions, such as high-altitude sites where maintaining good calibration is difficult. A significant limitation of this study is the lack of measurements in different atmospheric conditions. Measurements over a wide range of IWV would help significantly in strengthening the constraints on the parameters $a$ and $b$ in Eq. (5), particularly measurements where the continuum in the windows is dominated by either the foreign or self-continua. It is in principle possible to derive the absorption coefficients directly, given a set of atmospheric observations over a range of conditions, since the self-continuum varies with the square of the vapour pressure, while the foreign varies with the product of vapour pressure and the pressure of the ambient air. Such an analysis would also have to take into account the temperature dependence of the self-continuum.

If the relative contribution of the self and foreign-continua was well constrained, an analysis like that performed in Section 4 could be performed, but with significantly more confidence in the results, and allow a more direct comparison with the laboratory measurements without the strong assumptions required in our analysis.

Alternatively, one could use a horizontal atmospheric path, using e.g. a laser source rather than the Sun. This has been performed by e.g. Rieker et al., (2014) to observe carbon dioxide and methane absorption in the centre and edges of the 1.6 μm window using a frequency comb method over a 2 km path. Using a horizontal path reduces the effect of clouds and aerosols, and allows for *in situ* measurements of humidity, temperature and pressure directly in the beam path, rather than relying on potentially uncertain radiosonde measurements (which are directed by the prevailing winds and not necessarily representative of the path observed by a spectrometer). However, this would result in similar problems to those found in a laboratory, namely the difficulty in constructing a path length long enough to measure the comparatively weak continuum absorption in the windows. In such a measurement, the experimenter would have significantly less control over the conditions compared to a laboratory measurement.

##### 6. Conclusions

We have presented new field observations of the near-IR continuum in the atmospheric windows at 4, 2.1, 1.6, μm (2000-7000 cm$^{-1}$). These measurements are, to our knowledge, the first and only published measurements which characterise the water vapour continuum in the near-IR windows at sea-level. Our data show good agreement with laboratory spectra in the first two of these windows, but the agreement worsens with increasing wavenumber. This is consistent with signal contamination due to atmospheric aerosol, which is more pronounced at higher wavenumbers. These measurements provide some real-world validation of the extrapolated laboratory data and semi-empirical models, which are relied on for radiative modelling purposes.



In the centre of the 4 µm window, there is good agreement between the CAVIAR-field self-continuum and the various sets of laboratory data. The laboratory self-continua exhibit two different temperature dependencies, with Baranov and Lafferty (2011) showing a significantly steeper temperature dependence than Richard et al. (2017) .The CAVIAR-field data could agree with either of these implied temperature dependences, depending on whether the MT_CKD or CAVIAR-lab foreign continuum is assumed respectively. Given that CAVIAR-field is an experimental estimate, and the high precision and accuracy of the Richard et al. (2017) measurements, we believe that this is evidence for a weaker self-continuum at the centre of this window than observed by Baranov and Lafferty (2011). We also demonstrate that a strengthening is required to the MT_CKD foreign continuum in this window, in agreement with the results of Ptashnik et al. (2012), Baranov (2011) and Baranov and Lafferty (2012). This strengthening varies spectrally, but is a factor of ~100 in the centre of the window at 2500 cm$^{-1}$.

We show that, assuming the (temperature-extrapolated) CAVIAR-lab self-continuum is correct in the 2.1 µm window, the foreign continuum in the centre of the window is underestimated by MT_CKD by a factor of 5, in agreement with the laboratory measurements of Ptashnik et al., (2012) and Vasilchenko et al., (2019). In the centre of the window, assuming the CAVIAR-lab foreign continuum, our data agrees well with extrapolated self-continuum components from CAVIAR-lab and the various Grenoble CRDS measurements. At the edge of the window, we demonstrate that the MT_CKD foreign continuum is likely too weak, but by less than a factor of 5 and not as strong as the window-edge foreign continuum from CAVIAR-lab. Alternatively, there is a possibility that the foreign continuum exhibits more temperature dependence than has been inferred from the available laboratory studies.

At 1.6 µm, we show a significantly stronger implied self-continuum than the extrapolated CAVIAR-lab and Grenoble laboratory measurements, regardless of whether the MT_CKD or CAVIAR foreign continuum is used. This may indicate one of several things. There may be some systematic error in our retrieval of the continuum optical depth (whether due to aerosol, or a calibration issue). It may also suggest that a significantly stronger self-continuum is realistic, such as the large values reported by Ptashnik et al. (2015). However, this is unlikely to be the case, given that the authors of Ptashnik et al. (2015) believe their results may be spurious (Igor Ptashnik, pers. comm). Alternatively, it may be that the foreign continuum as measured by CAVIAR-lab may be too weak, or some combination of the above factors.

Across the spectrum, we observe a greater proportion of the total continuum optical depth in the 4, 2.1 and 1.6 µm windows as likely coming from the foreign continuum, rather than the self-continuum (for the atmospheric conditions at the time of our observations). This may indicate that the foreign continuum is being underestimated by MT_CKD in these windows, which could have significant implications for atmospheric radiative transfer calculations for both climate modelling and remote sensing applications.



Given the challenges that come with making absolutely calibrated high resolution results in the atmosphere, rather than a controlled laboratory setting, our results are characterised by high uncertainties. We detail ways in which a future field campaign should improve upon our characterisation of atmospheric aerosol in particular, by either mitigating its effect or measuring it with greater accuracy and precision.

This work represents a significant advance in understanding of the continuum absorption in near-IR windows, as it is the only existing dataset of direct atmospheric measurements with positive values in these windows. Our results are consistent with the upper limits imposed by Reichert and Sussmann, (2016). Our work, and that of Reichert and Sussmann, demonstrate that it is possible to observe the near-IR continuum in the field within the bands and windows to some degree of accuracy. We encourage

future field measurements, in as wide a range of conditions as possible, to more rigorously assess the partition between the self and foreign continua in the atmosphere. Such measurements should take steps to avoid the problems encountered in this work, particularly regarding aerosol scattering, with careful consideration of the calibration drift over the course of individual days of measurement and over the course of a measurement campaign.

## 7. Data availability

The data from this work are available via online depository (doi: 10.5281/zenodo.3520519). Included is the best estimate of 10 the continuum optical depth (18 September 2008), provided with the $k = 1$ uncertainties, the ratio of this best estimate to the corresponding MT_CKD 3.2 optical depth from 18 September 2008, and the CAVIAR-field estimated continuum absorption coefficients using both MT_CKD and CAVIAR-lab. All data are provided at 1 cm$^{-1}$ resolution. Other data (including full resolution data) are available from the corresponding author upon request.

**Acknowledgements**

CAVIAR (NE/D012082/1) was funded by the Natural Environment Research Council (NERC) and Engineering and Physical Sciences Research Council. Jonathan Elsey would like to thank the NERC SCENARIO Doctoral Training Partnership for funding his PhD research (studentship number 1503015), and NPL for providing additional financial support. We thank Liam Tallis for his work in performing the initial measurements. The latter stages of the analysis and writing were supported by the

Natural Environment Research Council "Advanced Spectroscopy for improved characterisation of the near-Infrared water vapour Continuum (ASPIC)" research grant (NE/R009848/1). We thank Nicolas Bellouin for his valuable input regarding the analysis of aerosol optical depth. We thank referees at the Access Review stage for many useful comments.



**Author contributions:**

JE led the analysis of the measurements, building on the initial work of KM, both of whom were supervised and guided by KS, MC and TG. MC and TG designed and calibrated the observing system and led the CAVIAR field measurements. KS was the overall Principal Investigator of the CAVIAR project and led the formulation of the project goals. JE led the writing of the original draft, with input from all other authors over several cycles of revision.

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
