# Peer review of "Atmospheric observations of the water vapour continuum in the nearinfrared windows between 2500-6600 cm-1"

_Atmospheric Measurement Techniques, 2019_

## Referee Comment (RC1) · Anonymous Referee #1 · 9 Jan 2020

Journal: AMTD Title: Atmospheric observations of the water vapour continuum in the near-infrared windows Author(s): Jonathan Elsey et al. MS No.: amt-2019-403

Summary & General comments This paper describes atmospheric measurements of the water vapor continuum in the near infrared windows (4.0, 2.1 and 1.6 $\mu$m) from spectra recorded with a radiometrically-calibrated Fourier transform spectrometer. The optical depth, due to water continuum (self+foreign), was retrieved after subtraction of the line-by-line, Rayleigh and aerosols contributions to the total optical depth derived with the Langley method. The optical depth due to the self-continuum is then obtained assuming either the MT_CKD3.2 foreign continuum or the CAVIAR laboratory measurements of the foreign-continuum with no temperature dependence. Vice versa, the optical depth due to the foreign-continuum is obtained assuming either the MT_CKD3.2 self-continuum or the CAVIAR laboratory measurements of self-continuum extrapolated at 280 K. Both (self- and foreign-) continua are compared to the existing literature data.

The paper is very-well written and almost all the necessary details are given. The work reported here is of high quality. This paper also shows and discusses the current limitations of the atmospheric determination of the water vapor continuum due to potential biases coming from the aerosols, mirror reflectivity and phase corrections. Even if uncertainties on the continuum optical depths are still large, this kind of study is important to test existing laboratory data. Ideas to reduce these uncertainties on retrieved continuum are discussed at the end of the paper. The paper is fully in the scope of AMT and is well-suited for a publication in this journal after some minor corrections (see the comments and remarks below).

Main remarks/comments: P3, 1st paragraph: The first sentence "Measuring the continuum...present in the atmosphere" is too general. For example CRDS/OF-CEAS techniques allow for measurements at room temperature and at low pressure close to atmospheric conditions. Same remark for the next sentences: equivalent pathlenght with CRDS/OF-CEAS techniques can reach several hundreds of km and base lines are highly stable.

P6, L7: The authors have to specified the cutoff value for the Voigt profile and if they include or not in the continuum the plinth below the absorption lines.

P9, Figure 3: On panel (d) the water vapour optical depth is around 0.025. This value doesn't correspond to values reported in Fig 4 and in Supplementary Material which are between 0.01 and 0.008 for the same spectral region. Can the authors clarify this?

P9 or P10: In addition to figures 4 and 5, a figure showing the relative contribution of the aerosols and of the continuum to the optical depth after subtraction of the line-byline and Rayleigh contributions will be very helpful to demonstrate the importance of the aerosols optical depth knowledge.

P19, L17-18: . . .due to the lack of laboratory measurements at atmospheric temperatures, one must assume a temperature dependence of the self-continuum. This sentence has to be reformulated as CRDS/OF-CEAS measurements of the self-continuum are available at room temperature. Why the authors did not adopt this data set instead of the extrapolated high temperature data of the CAVIAR laboratory measurements?

P21, L25-29: In these lines, authors discuss the two possible temperature dependences and they seem to have the same "degree of confidence" in both. This is a little bit strange as they decided to replace the room temperature CAVIAR data by the values extrapolated at 280 K from high temperature CAVIAR measurements.

P24, Fig 14 (b): The data point called Mondelain et al. (2015) should not be plotted on this panel as it was obtained at 4250 cm-1 and not at 4300 cm-1.

P24, L20: The authors should mention here that the difference is due to the fact that, in one case (Vasilchenko et al) a purely quadratic function was used to fit the data considering that there was no adsorption on the mirrors and that in Mondelain et al an additional linear term was used to take into account the supposed adsorption contribution.

P26, L5: The authors should mention that in the 4 $\mu$m window the continuum is stronger than MT_CKD and the extrapolated CAVIAR-lab self-continuum.

P26, L9: The authors should add: . . . a factor of 100 would be required to bring the CAVIAR-lab and CAVIAR-field self-continua into agreement, in contradiction with CAVIAR foreign continuum.

P27, Figure 16: Several experimental points from CRDS/OF-CEAS experiments are missing in the 4 $\mu$m window (see Campargue 2016 and Richard 2017) and in the 2.1 $\mu$m window. In the 1.6 $\mu$m window the plotted data have to be replaced by the more

recent measurements of Vasilchenko 2019.

In Figure 16 (and also in Fig. 17), the uncertainties on the CAVIAR lab measurements are missing and have to be added.

P28, L 29-30: A strong affirmation is made here by considering that almost all the continuum observed in Reichert and Sussmann is due to the foreign contribution. The authors have to justify this.

Specific comments P3, L12: ...the adjustement of the water vapour...

P3, L13: in addition to additional empirical adjustments?

P3, L32: Only the reference for the foreign-continuum is given. Which self-continuum cross-section is used to obtain the estimated values given in the sentence?

P6, Eq. (2): What means $\tau$other as there is already $\tau$other_gases in the equation?

P18, L13: ...self-continuum cross-section...

P35, L17: The term AOD has to be defined.

P36, L7: Such an analysis...

P36, L25: ... the water vapour self-continuum in the near-IR windows at sea level.

In AMT paper supplementary: P2: Just before equation (S12) it is written y=mx+c. This is misleading as in fact x equal to m in Equation (S12). Authors should replace m by b for example.

P3: Additionally, the agreement between the Langley and closure data (Figure 8)...

Figure S3: Cosinus is missing in the legend of the y-axis. Moreover the angle $\theta$ is already used at the beginning of the paper to name the solar zenith angle. Another Greek letter should be used.

Please also note the supplement to this comment:

https://www.atmos-meas-tech-discuss.net/amt-2019-403/amt-2019-403-RC1-supplement.pdf

---

## Referee Comment (RC2) · Anonymous Referee #2 · 24 Jan 2020

General comments

This discussion paper deals with the determination of the water vapor continuum using observations of a radiometrically calibrated Fourier transform spectrometer. The continuum optical depth retrieval considers all relevant contributions ranging from aerosol optical depth over Rayleigh and spectral line contributions to other continua and thus fits well within the scope of AMT. The data used and presented are only few (from one day) but new. By a comparison of their results to the current MT_CKD continuum model the authors come to substantial conclusions that could be used for the further improvement of MT_CKD.

[Figure]

The used Langley method is scientifically valid and the results are mostly sufficient to support the interpretations. In general, the paper is well written and the authors give proper credit to related work. The authors manage a comprehensive comparison to other laboratory and field observations. The title reflects the contents of the paper and the abstract provides a concise summary. Smaller suggestions for improvement are given in the next section with the specific comments.

A small adaption of the overall structure could improve the readability of the paper further. Section 3 with the results contains with subsection 3.2 a comparison with MT_CKD. Section 4 is then about the comparison with laboratory observations.

Although section 3.2 is about optical depth and section 4 mostly about cross sections, it could be an advantage to lift both to the same level of sections.

Specific comments

P3, line 13: The authors state that "in many cases they use either version 2.5 or version 3.2" of MT_CKD. It would be helpful, if the authors could give a more specific reference or a short indicative list of some relevant cases.

P4, line 22: How do the authors come to the conclusion that Zugspitze measurements were taken at airmass factors of ∼6? Please explain this in more detail.

P5 concerning experimental setup: It remains unclear how the Microtops II sunphotometer was operated. Was this handheld device mounted on a stand/tripod? Was it mounted on a solar tracker? It would be helpful if the authors describe how it was ensured that the aerosol optical depth measurements were performed along the same atmospheric path.

P9, fig. 3c: The data shown is marked as smoothed. How exactly was this smoothing mathematically performed? Is it the same smoothing about 15 cm-1 mentioned for the continuum on p6, line 14?

P10, fig. 4: In this figure the blue shading corresponds to k=1 and the cyan shading

to k=2 uncertainty. There seems to be an envelope below and above the cyan shading that is colored again blue. If there is a physical meaning of it, could the authors please explain it?

P13, line 8: A suggestion for improvement is to mention the magnitude of the field-of-views of both the Microtops and the FTS.

P16, fig. 8: Is there any reason why the Langley and closure method derived optical depths in the upper part of the figure do not cover the same region of the residual in the lower part of the figure? If possible, they should be the same.

P19, line 4: The section title with laboratory observations fits to the lab measurements, but does not quite fit to the comparison with Reichert and Sussmann (2016) that are also included in the comparison. Their observations were field observations as the CAVIAR field data in this paper.

P19, line 8: The derived continuum optical depth tau_totalˆCAV has another naming in the following formulas, e.g. formulas (3) and (5). Additionally, the quantity tau_forˆlab mentioned on P20, fig. 11 was not introduced.

P24, line 10: The authors refer to lower temperature data (cyan point and dashed line), but in figure 13 there is no cyan point and no cyan dashed line. Seemingly, this passage is from an earlier version of this paper. CAVIAR-lab (297K) should be removed from the legend in figure 13.

P26, fig. 14: CAVIAR-lab (297) is not anymore included in the figure, so it should be removed from the legend. The same applies to the caption.

P29, fig. 16: At the beginning of the second line of the caption self-continuum is assumed to be the foreign-continuum.

P33, fig. 17: In the caption it would be more precise, to mention that the showed data corresponds only to atmospheric windows in the mentioned region. The authors could insert "in atmospheric windows" between continuum and across.

P33, fig. 17: Concerning the showed Reichert and Sussmann (2016) data, ignored is the fact that they used MT_CKD_2.5.2 model for their continuum retrieval. As the self-continuum was assumed to be consistent with the MT_CKD model a direct comparison like in this figure is challenging.

P38, line 27: Constraining the spectral coverage from 2000-7000 cm-1 to 2100-6600 cm-1 would be more precisely.

Supplement: The airmass factor definition m = cos teta contradicts the airmass factor definition given in the paper. The Beer-Bouguer-Lambert law given here is only valid with m = 1/cos teta.

Technical corrections

P3, line 7: remove "at" after temperature

P10, line 5: word repetition (distance 2) of approximation/approximately

P12, line 9: insert vapor (or vapour) between water and continuum

P27, line 3: Period/full stop is missing right after "term".

P37, line 27, word repetition (distance 1) of aircraft

P40, line 16: remove "10", which seems to be a line number of an earlier version of this paper

Supplement, P2: remove "the" in front of "account" in the third-to-last paragraph

There is an inconsistency in the writing of the MT_CKD versions. Mostly the current version is named MT_CKD3.2, but sometimes the naming is with a space in front of the version number. For the future reader a coherent way of writing would be an advantage, e.g. in browsing the paper. The model's developers are using with MT_CKD_3.2 a third way of spelling.

---

## Referee Comment (RC3) · Anonymous Referee #2 · 10 Feb 2020

Page 33, Fig. 17: The authors used the HITRAN2016 spectroscopic database for their line-by-line absorption calculations. However, the retrieval of Reichert and Sussmann (2016) that is included in the figure is based on aer_v_3.2, that is provided alongside the LBLRTM model. This line list consists mainly of HITRAN2008. Did the authors consider, that a quantitative description of the water vapor continuum is related to the set of line parameter values used? This should at least be dicussed.

---

## Author Comment (AC1) · 10 Feb 2020

Authors' response to reviewer comments

*We thank both reviewers for their time and for their many suggestions/comments, which will help improve the manuscript. We detail the responses to each of these comments in turn below.*

*We will also take the opportunity to improve the flow of our manuscript by dividing Section 2 into various sub-sections, and change some of the terminology used for consistency.*

**Reviewer #1**

*We note that many of these issues have been addressed prior to publication in AMT Discussions following comments made by the same reviewer during the Technical Corrections stage. Our responses to these comments reflect the changes that were made to the manuscript at this stage, and make explicit where any further changes have been made.*

**Main remarks/comments:**

P3, 1st paragraph: The first sentence "Measuring the continuum…present in the atmosphere" is too general. For example CRDS/OF-CEAS techniques allow for measurements at room temperature and at low pressure close to atmospheric conditions. Same remark for the next sentences: equivalent pathlenght with CRDS/OF-CEAS techniques can reach several hundreds of km and base lines are highly stable.

*We agree that CRDS allows for measurements at room temperature (and in principle lower) and at low pressure, and have added in a statement to this effect. Room temperature is not the same as atmospheric temperature however; there are not yet (to our knowledge) measurements by CRDS of the continuum in these windows at temperatures as low as 280K, which are the atmospherically relevant temperatures we refer to here. We have changed the text to indicate that we consider temperatures below room temperature and added a further sentence to acknowledge the importance of CRDS at room temperature.*

P6, L7: The authors have to specified the cutoff value for the Voigt profile and if they include or not in the continuum the plinth below the absorption lines.

*This is the standard 25 cm$^{-1}$ with the plinth subtracted from the absorption lines, as it is assumed to be included in the continuum. We have now explicitly included this information in the manuscript.*

P9, Figure 3: On panel (d) the water vapour optical depth is around 0.025. This value doesn't correspond to values reported in Fig 4 and in Supplementary Material which are between 0.01 and m0.008 for the same spectral region. Can the authors clarify this?

*Thank you for spotting this. This Figure was in error, due to a software bug, and has now been corrected.*

P9 or P10: In addition to figures 4 and 5, a figure showing the relative contribution of the aerosols and of the continuum to the optical depth after subtraction of the line-by-line and Rayleigh contributions will be very helpful to demonstrate the importance of the aerosols optical depth knowledge.

*We agree. Figure 5 has been updated with a second panel showing the relative contribution of aerosol and continuum to the combined continuum + aerosol optical depth.*

P19, L17-18: …due to the lack of laboratory measurements at atmospheric temperatures, one must assume a temperature dependence of the self-continuum. This sentence has to be reformulated as CRDS/OF-CEAS measurements of the self-continuum are available at room temperature. Why the authors did not adopt this data set instead of the extrapolated high temperature data of the CAVIAR laboratory measurements?

*Even with room-temperature measurements of self-continuum, it is still necessary to extrapolate down to ~280K for our purposes. We have made this more explicit in the manuscript. We chose to use the CAVIAR-lab data since it has broad spectral coverage across all the windows, and measurements at a range of temperatures across each of these windows.*

P21, L25-29: In these lines, authors discuss the two possible temperature dependences and they seem to have the same "degree of confidence" in both. This is a little bit strange as they decided to replace the room temperature CAVIAR data by the values extrapolated at 280 K from high temperature CAVIAR measurements.

*Our stance on this is that there are two possible temperature dependences; given the consistency of the straight line fit through the high-T CAVIAR-lab data, we believe it is reasonable to suggest that there is possibly an issue with the low-T CAVIAR-lab data in this window, unless there is some unexpected temperature dependence. This lower temperature dependence is also consistent with the OF-CEAS data of Richard et al. It is our belief that this is likely to be a more robust estimate of the temperature dependence given the agreement between the high-T CAVIAR-lab data and the Richard et al. data. This paragraph has been reworded, and the figure caption updated with the low-temperature CAVIAR-lab data point, as we agree that it was not clear.*

P24, Fig 14 (b): The data point called Mondelain et al. (2015) should not be plotted on this panel as it was obtained at 4250 cm-1 and not at 4300 cm-1.

*Thank you for pointing out this error. This data point has been removed from the Figure.*

P24, L20: The authors should mention here that the difference is due to the fact that, in one case (Vasilchenko et al) a purely quadratic function was used to fit the data considering that there was no adsorption on the mirrors and that in Mondelain et al an additional linear term was used to take into account the supposed adsorption contribution.

*A sentence has been added to this effect explaining why the two do not agree within the uncertainties.*

P26, L5: The authors should mention that in the 4 μm window the continuum is stronger than MT_CKD and the extrapolated CAVIAR-lab self-continuum.

*This has been added.*

P26, L9: The authors should add: … a factor of 100 would be required to bring the CAVIAR-lab and CAVIAR-field self-continua into agreement, in contradiction with CAVIAR foreign continuum.

*We agree that this is inconsistent with the majority of available data, and have included this rewording into the sentence.*

P27, Figure 16: Several experimental points from CRDS/OF-CEAS experiments are missing in the 4 μm window (see Campargue 2016 and Richard 2017) and in the 2.1 μm window. In the 1.6 μm window the plotted data have to be replaced by the more recent measurements of Vasilchenko 2019. In Figure 16 (and also in Fig. 17), the uncertainties on the CAVIAR lab measurements are missing and have to be added.

*These data points have been added to the manuscript. We have added the uncertainties in the CAVIAR-lab measurements to the plots; in the self-continuum case we scaled the uncertainties from the higher-temperature measurements using the same extrapolation. We also make clear that the comparison here is with the various Grenoble measurements at room temperature, and indicate that an additional scaling factor would need to be applied to these measurements to bring them down to ~280 K.*

*The next revision of the paper will include an updated uncertainty budget, with an additional term due to this extrapolation in temperature to 280 K.*

P28, L 29-30: A strong affirmation is made here by considering that almost all the continuum observed in Reichert and Sussmann is due to the foreign contribution. The authors have to justify this.

*We justify this based on Page 9 lines 2-4 of Reichert and Sussmann (2016), which states "the foreign continuum… is by far dominant for most spectral regions given the dry atmospheric conditions encountered in [their] data set.". We have added extra sentences to the text to reflect this.*

**Specific comments**

P3, L12: …the adjustement of the water vapour…

*This has been corrected.*

P3, L13: in addition to additional empirical adjustments?

*This has been reworded to avoid the repetition.*

P3, L32: Only the reference for the foreign-continuum is given. Which self-continuum cross-section is used to obtain the estimated values given in the sentence?

*The given reference (Ptashnik et al. (2012)) contains within it the estimates for the partitioning of the foreign and self-continuum based on Ptashnik et al. (2011a) and Ptashnik et al. (2012). The sentence has been updated to reflect this.*

P6, Eq. (2): What means $\tau_{other}$ as there is already $\tau_{other\_gases}$ in the equation?

*This refers to other continua, such as the $O_2$ and $CO_2$ continua which are included in MT_CKD. The text now more explicitly refers to this.*

P18, L13: …self-continuum cross-section…

*A "c" has been added to the beginning of cross-section to fix this.*

P35, L17: The term AOD has to be defined.

*The acronym "AOD" has been removed and replaced with $\tau_{aerosol}$, to remain consistent with the rest of the manuscript.*

P36, L7: Such an analysis…

*The extraneous "a" has been removed from this sentence.*

P36, L25: … the water vapour self-continuum in the near-IR windows at sea level.

*The hyphen has been removed.*

**In AMT paper supplementary:**

P2: Just before equation (S12) it is written y=mx+c. This is misleading as in fact x equal to m in Equation (S12). Authors should replace m by b for example.

*Thank you for pointing this out. We have changed the format of the linear equation from y = mx + c to y = ax + b, to remove this confusion.*

P3: Additionally, the agreement between the Langley and closure data (Figure 8)…

*This has been changed.*

Figure S3: Cosinus is missing in the legend of the y-axis. Moreover the angle θ is already used at the beginning of the paper to name the solar zenith angle. Another Greek letter should be used.

*The cosine has been added and theta replaced with phi, to avoid this confusion.*

**Reviewer #2**

A small adaption of the overall structure could improve the readability of the paper further. Section 3 with the results contains with subsection 3.2 a comparison with MT_CKD. Section 4 is then about the comparison with laboratory observations. Although section 3.2 is about optical depth and section 4 mostly about cross sections, it could be an advantage to lift both to the same level of sections.

*We agree that the treatment of MT_CKD and the laboratory observations is inconsistent. We will therefore change Section 3.2 such that it is now its own Section (section 4).*

Specific comments

P3, line 13: The authors state that "in many cases they use either version 2.5 or version3.2" of MT_CKD. It would be helpful, if the authors could give a more specific reference or a short indicative list of some relevant cases.

*We agree that including some examples with citations is beneficial here; we will include a list of various codes which use MT_CKD, mentioning the version number where made explicit in the references.*

P4, line 22: How do the authors come to the conclusion that Zugspitze measurements were taken at airmass factors of ~6? Please explain this in more detail.

*This number is taken from the fact that the lowest airmass factor at the surface for Zugspitze in Dec/Jan is ~3, and the limit imposed by the authors indicating that only observations with airmass below 9 are used. The sentence will be changed to reflect this range of airmasses more precisely.*

P5 concerning experimental setup: It remains unclear how the Microtops II sunphotometer was operated. Was this handheld device mounted on a stand/tripod? Was it mounted on a solar tracker? It would be helpful if the authors describe how it was ensured that the aerosol optical depth measurements were performed along the same atmospheric path.

*The Microtops sunphotometer was operated by hand by an operator at the field site co-incident with the FTIR measurements – the lack of a dedicated solar tracker on this is potentially a source of uncertainty in the aerosol measurements. However, this is not something that we can quantify or cross-check, given the lack of measurements from another source other than the FTIR. We will include this caveat when discussing the Microtops measurements, and will add the need for better tracking of the solar disc as a suggestion in Section 5.*

P9, fig. 3c: The data shown is marked as smoothed. How exactly was this smoothing mathematically performed? Is it the same smoothing about 15 cm-1 mentioned for the continuum on p6, line 14?

*This is the same smoothing (at 15 cm$^{-1}$) as performed for the final analysis, which was performed using a moving average (boxcar) filter. The text will be updated to reflect this.*

P10, fig. 4: In this figure the blue shading corresponds to k=1 and the cyan shading to k=2 uncertainty. There seems to be an envelope below and above the cyan shading that is colored again blue. If there is a physical meaning of it, could the authors please explain it?

*These lines are there to demarcate the edges of the uncertainty limits. Given that they caused ambiguity as to their meaning, they will be removed.*

P13, line 8: A suggestion for improvement is to mention the magnitude of the field-of-views of both the Microtops and the FTS.

*We will now include the FOV of the Microtops and FTS within the manuscript, and point toward our discussion of the forward scattering/FOV issue on Page 13.*

P16, fig. 8: Is there any reason why the Langley and closure method derived optical depths in the upper part of the figure do not cover the same region of the residual in the lower part of the figure? If possible, they should be the same

*We have now updated the figure to show the residual on the same scale as the top panel.*

.P19, line 4: The section title with laboratory observations fits to the lab measurements, but does not quite fit to the comparison with Reichert and Sussmann (2016) that are also included in the comparison.  Their observations were field observations as the CAVIAR field data in this paper.

*Section 4 (now Section 5, see top-level response to Reviewer #2 above) has been renamed to "Comparison with other observations", and the text in the first paragraph changed to reflect that we are also comparing to these field observations.*

P19, line 8: The derived continuum optical depth tau_total^CAV has another naming in the following formulas, e.g. formulas (3) and (5).  Additionally, the quantity tau_for^lab mentioned on P20, fig. 11 was not introduced

*Tau_total^CAV will be renamed to more accurately fit what is in the Figures and the Equations. Equation (4) will be updated to fix a typo, where the left-hand-side of the equation is equal to tau_for^CAV rather than the correct tau_for^lab.*

.P24, line 10: The authors refer to lower temperature data (cyan point and dashed line), but in figure 13 there is no cyan point and no cyan dashed line. Seemingly, this passage is from an earlier version of this paper.  CAVIAR-lab (297K) should be removed from the legend in figure 13.

*The version of Figures 13 and 14 used in the uploaded drafts of the paper do not include this data point and the corresponding extrapolation – this was in error. The correct versions of these*

*Figures are shown below – they are the figures uploaded for the original draft, but including the 297 K data in the legend. These will be included for the final version.*

[Figure]

P26, fig. 14: CAVIAR-lab (297) is not anymore included in the figure, so it should be removed from the legend. The same applies to the caption.

*See previous comment.*

P29, fig. 16: At the beginning of the second line of the caption self-continuum is assumed to be the foreign-continuum.

*The caption will be fixed to reflect that the CAVIAR-field self-continuum is estimated using the CAVIAR-lab foreign-continuum.*

P33, fig. 17: In the caption it would be more precise, to mention that the showed data corresponds only to atmospheric windows in the mentioned region. The authors could insert "in atmospheric windows" between continuum and across.

*We will now make explicit that the CAVIAR-field data corresponds to the window regions only, and use "CAVIAR-field" rather than "Langley-estimated" for consistency.*

P33, fig. 17: Concerning the showed Reichert and Sussmann (2016) data, ignored is the fact that they used MT_CKD_2.5.2 model for their continuum retrieval. As the self-continuum was assumed to be consistent with the MT_CKD model a direct comparison like in this figure is challenging.

*We believe that this comparison is reasonable, given we have used MT_CKD_3.2 in panel a) of this Figure to obtain the foreign continuum, and that since the foreign continuum contribution is dominant in the Reichert and Sussmann (2016) case, we believe that it is not likely to have an impactful effect on the comparison. We will add an additional sentence to the Figure caption and to the text to make clear that this is the case.*

P38, line 27: Constraining the spectral coverage from 2000-7000 cm-1 to 2100-6600cm-1 would be more precisely .

*The text will be updated to match the title and more precisely reflect the wavenumber range.*

Supplement: The airmass factor definition m = cos teta contradicts the airmass factor definition given in the paper. The Beer-Bouguer-Lambert law given here is only valid with m = 1/cos teta.

*This was a typo and will be corrected.*

Technical corrections

P3, line 7: remove "at" after temperature

*This will be corrected to "as".*

P10, line 5: word repetition (distance 2) of approximation/approximately

*This sentence will be reworded to "…which in the limit of small absorption is approximately the optical depth noise in that region".*

P12, line 9: insert vapor (or vapour) between water and continuum

*This will be added.*

P27, line 3: Period/full stop is missing right after "term".

*This full stop will be added.*

P37, line 27, word repetition (distance 1) of aircraft

*We will now reword this to remove the repetition, and also define FAAM.*

P40, line 16: remove "10", which seems to be a line number of an earlier version of this paper

*This will be removed.*

Supplement, P2: remove "the" in front of "account" in the third-to-last paragraph

*This will be removed.*

There is an inconsistency in the writing of the MT_CKD versions.  Mostly the current version is named MT_CKD3.2, but sometimes the naming is with a space in front of the version number. For the future reader a coherent way of writing would be an advantage, e.g. in browsing the paper. The model's developers are using with MT_CKD_3.2 a third way of spelling

*We shall ensure that all references to MT_CKD now use the MT_CKD_ syntax, to be consistent with other literature.*

---

## Author Comment (AC3) · 19 Feb 2020

Page 33, Fig. 17: The authors used the HITRAN2016 spectroscopic database for their line-by-line absorption calculations. However, the retrieval of Reichert and Sussmann (2016) that is included in the figure is based on aer_v_3.2, that is provided alongside the LBLRTM model. This line list consists mainly of HITRAN2008. Did the authors consider, that a quantitative description of the water vapor continuum is related to the set of line parameter values used? This should at least be dicussed.

*We agree with the Reviewer's comment. Given the standard definition of the continuum as including the contribution from spectral lines outside 25 cm$^{-1}$ from line centre (and their "pedestal" within 25 cm$^{-1}$) , any attribution of the continuum will depend to some extent on the linelist used. We will therefore include some discussion of this issue within the paper, which is of potential importance for our comparisons with laboratory derivations of the continuum, as well as with Reichert and Sussmann (2016).*

*The impact of the change between HITRAN2008 and HITRAN2016 on our derived continuum absorption is small in the 2.1 and 1.6 µm windows (see attached Figures, which show the optical depth for the main observational day (18 Sept 2008) in our paper), with a more significant effect in the centre of the 4 µm window. Typically, in the centre of the windows the derived continuum optical depth is of order 0.01. In the centre of the 1.6 and 2.1 µm windows, the choice of HITRAN database affects this derived continuum by less than 20%. In the 4 µm window, the impact reaches 80% at some wavenumbers in the centre of the window, but the effect is around 40% at most wavenumbers. We note however that the advances in laboratory and theoretical studies between the releases of HITRAN2008 and HITRAN2016 indicate that the latter is more likely to be accurate, which gives a slightly weakened continuum in the centre of the 4 µm window compared to HITRAN2008.*

*HITRAN 2016:*

---

## Author Response (AR2)

Authors' response to reviewer comments

*We thank both reviewers for their time and for their many suggestions/comments, which have improved the manuscript. We detail our resposnes to each of these comments in turn below, followed by the changes made in the manuscript.*

*We have also taken the opportunity to improve the flow of our manuscript by dividing Section 2 into various sub-sections, and change some of the terminology used for consistency where there were previously differences. These changes can be seen in the Tracked Changes version of the document.*

*Where line numbers are mentioned in our responses, these correspond to the line numbers of the revised document unless otherwise stated.*

**Reviewer #1**

*We note that many of these issues have been addressed prior to publication in AMT Discussions following comments made by the same reviewer during the Technical Corrections stage. Our responses to these comments reflect the changes that were made to the manuscript at this stage, and make explicit where any further changes have been made.*

**Main remarks/comments:**

P3, 1st paragraph: The first sentence "Measuring the continuum…present in the atmosphere" is too general. For example CRDS/OF-CEAS techniques allow for measurements at room temperature and at low pressure close to atmospheric conditions. Same remark for the next sentences: equivalent pathlenght with CRDS/OF-CEAS techniques can reach several hundreds of km and base lines are highly stable.

*We agree that CRDS allows for measurements at room temperature (and in principle lower) and at low pressure, and have added in a statement to this effect. Room temperature is not the same as atmospheric temperature however; there are not yet (to our knowledge) measurements by CRDS of the continuum in these windows at temperatures as low as 280K, which are the atmospherically relevant temperatures we refer to here. We have changed the text (P3, L5) to indicate that we consider temperatures below room temperature and added a further sentence to acknowledge the importance of CRDS at room temperature.*

P6, L7: The authors have to specified the cutoff value for the Voigt profile and if they include or not in the continuum the plinth below the absorption lines.

*This is the standard 25 $cm^{-1}$ with the plinth subtracted from the absorption lines, as it is assumed to be included in the continuum. We have now explicitly included this information in the manuscript on P6 L13.*

P9, Figure 3: On panel (d) the water vapour optical depth is around 0.025. This value doesn't correspond to values reported in Fig 4 and in Supplementary Material which are between 0.01 and m0.008 for the same spectral region. Can the authors clarify this?

*Thank you for spotting this. This Figure was in error, due to a software bug, and has now been corrected.*

P9 or P10: In addition to figures 4 and 5, a figure showing the relative contribution of the aerosols and of the continuum to the optical depth after subtraction of the line-by-line and Rayleigh contributions will be very helpful to demonstrate the importance of the aerosols optical depth knowledge.

*We agree. Figure 5 has been updated with a second panel showing the relative contribution of aerosol and continuum to the combined continuum + aerosol optical depth.*

P19, L17-18: …due to the lack of laboratory measurements at atmospheric temperatures, one must assume a temperature dependence of the self-continuum. This sentence has to be reformulated as CRDS/OF-CEAS measurements of the self-continuum are available at room temperature. Why the authors did not adopt this data set instead of the extrapolated high temperature data of the CAVIAR laboratory measurements?

*Even with room-temperature measurements of self-continuum, it is still necessary to extrapolate down to ~280K for our purposes. We chose to use the CAVIAR-lab data since it has broad spectral coverage across all the windows, and measurements at a range of temperatures across each of these windows. We have made this more explicit in the manuscript (P21 L5).*

P21, L25-29: In these lines, authors discuss the two possible temperature dependences and they seem to have the same "degree of confidence" in both. This is a little bit strange as they decided to replace the room temperature CAVIAR data by the values extrapolated at 280 K from high temperature CAVIAR measurements.

*Our stance on this is that there are two possible temperature dependences; given the consistency of the straight line fit through the high-T CAVIAR-lab data, we believe it is reasonable to suggest that there is possibly an issue with the low-T CAVIAR-lab data in this window, unless there is some unexpected temperature dependence. This lower temperature dependence is also consistent with the OF-CEAS data of Richard et al. It is our belief that this is likely to be a more robust estimate of the temperature dependence given the agreement between the high-T CAVIAR-lab data and the Richard et al. data.*

*The paragraph has been modified starting on P23 L2 to say "There are therefore two possible experimentally-implied temperature dependences, a lower one implied by the high-temperature CAVIAR-lab, Richard et al. (2017) and Burch and Alt (1984) measurements, and a stronger dependence implied by the less-certain lower-temperature CAVIAR-lab, and Baranov and Lafferty (2011) measurements.", and the figure caption updated with the low-temperature CAVIAR-lab data point, as we agree that it was not clear.*

P24, Fig 14 (b): The data point called Mondelain et al. (2015) should not be plotted on this panel as it was obtained at 4250 cm-1 and not at 4300 cm-1.

*Thank you for pointing out this error. This data point has been removed from the Figure.*

P24, L20: The authors should mention here that the difference is due to the fact that, in one case (Vasilchenko et al) a purely quadratic function was used to fit the data considering that there was no adsorption on the mirrors and that in Mondelain et al an additional linear term was used to take into account the supposed adsorption contribution.

*A sentence has been added to this effect explaining why the two do not agree within the uncertainties (P27 L5), saying "This [difference] can be attributed to the differences in the fits used to obtain these cross-sections; both used a quadratic fit of vapour pressure versus absorption to obtain their cross-sections, but Mondelain et al. use an additional linear term to account for supposed adsorption on the mirrors, whereas Vasilchenko et al. did not need to use this additional term. The Vasilchenko et al., (2019) data, being more recent, is regarded as the more reliable, but is only available at one temperature."*

P26, L5: The authors should mention that in the 4 μm window the continuum is stronger than MT_CKD and the extrapolated CAVIAR-lab self-continuum.

*We agree that this should be made explicit. This has been added; P28 L11 now states: "However, there are clearly some issues in the 4 μm window, where at lower wavenumbers the derived continuum is significantly stronger than MT_CKD_3.2 and the extrapolated CAVIAR-lab self-continuum, and in the low-wavenumber edge of the 2.1 μm window, where when estimating the self-continuum using the CAVIAR-lab foreign continuum there is a significant decrease in the self-continuum."*

P26, L9: The authors should add: … a factor of 100 would be required to bring the CAVIAR-lab and CAVIAR-field self-continua into agreement, in contradiction with CAVIAR foreign continuum.

*We agree that this is inconsistent with the majority of available data, and have included this rewording into the sentence. P28 L16 now states "In the 1.6 μm window, a significant strengthening of the foreign continuum of over a factor of 100 would be required to bring the central values of the CAVIAR-lab and CAVIAR-field self-continua into agreement, which is inconsistent with the CAVIAR-lab foreign continuum in this window".*

P27, Figure 16: Several experimental points from CRDS/OF-CEAS experiments are missing in the 4 μm window (see Campargue 2016 and Richard 2017) and in the 2.1 μm window. In the 1.6 μm window the plotted data have to be replaced by the more recent measurements of Vasilchenko 2019. In Figure 16 (and also in Fig. 17), the uncertainties on the CAVIAR lab measurements are missing and have to be added.

*These data points have been added to the Figures. We have added the uncertainties in the CAVIAR-lab measurements to the plots; in the self-continuum case we used a Monte Carlo simulation to extrapolate the uncertainty in the 280 K data from the higher-temperature data, which takes into account the uncertainty in these data and uncertainty in the extrapolation. We also make clear that the comparison here is with the various Grenoble measurements at room temperature, and indicate that an additional scaling factor would need to be applied to these measurements to bring them down to ~280 K.*

*P28 L8 now states "The CRDS measurements are shown at their original temperature, since for many wavenumbers observations are only available at one temperature. Assuming the MT_CKD temperature dependence holds, these can be scaled by a factor of ~1.35 to be brought to 280 K."*

P28, L 29-30: A strong affirmation is made here by considering that almost all the continuum observed in Reichert and Sussmann is due to the foreign contribution. The authors have to justify this.

*We justify this based on Page 9 lines 2-4 of Reichert and Sussmann (2016), which states "the foreign continuum… is by far dominant for most spectral regions given the dry atmospheric conditions encountered in [their] data set.".*

*P30 L31 now states "Given the high altitude and low water vapour path of their measurements, Reichert and Sussmann indicate that the foreign continuum is by far the dominant contributor to the continuum in the majority of their measured spectral regions…"*

**Specific comments**

P3, L12: …the adjusement of the water vapour…

*P3 L17 has been corrected to read "In the window regions, the MT_CKD continuum mostly originates from adjustment of the water vapour lineshape…"*

P3, L13: in addition to additional empirical adjustments?

*P3 L19 now says "with additional empirical adjustments" to avoid the repetition.*

P3, L32: Only the reference for the foreign-continuum is given. Which self-continuum cross-section is used to obtain the estimated values given in the sentence?

*The given reference (Ptashnik et al. (2012)) contains within it the estimates for the partitioning of the foreign and self-continuum based on Ptashnik et al. (2011a) and Ptashnik et al. (2012). The sentence has been updated to reflect this; P4 L3 now states "(as calculated in Ptashnik et al., 2012)"*

P6, Eq. (2): What means τother as there is already τother_gases in the equation?

*This refers to other continua, such as the O2 and CO2 continua which are included in MT_CKD. The text now more explicitly refers to this. P6 L30 now reads "Continuum absorption by other molecules (N2, O2, O3 and CO2, defined here as τ_other...)"*

P18, L13: …self-continuum cross-section…

*A "c" has been added to the beginning of cross-section to fix this.*

P35, L17: The term AOD has to be defined.

*The acronym "AOD" has been removed from all instances of the manuscript and replaced with $\tau_{aerosol}$, to remain consistent with the rest of the manuscript.*

P36, L7: Such an analysis…

*The extraneous "a" has been removed from this sentence.*

P36, L25: … the water vapour self-continuum in the near-IR windows at sea level.

*The hyphen has been removed.*

**In AMT paper supplementary:**

P2: Just before equation (S12) it is written y=mx+c. This is misleading as in fact x equal to m in Equation (S12). Authors should replace m by b for example.

*Thank you for pointing this out. We have changed the format of the linear equation from y = mx + c to y = ax + b, to remove this confusion.*

P3: Additionally, the agreement between the Langley and closure data (Figure 8)…

*This has been changed.*

Figure S3: Cosinus is missing in the legend of the y-axis. Moreover the angle θ is already used at the beginning of the paper to name the solar zenith angle. Another Greek letter should be used.

*The cosine has been added and theta replaced with phi, to avoid this confusion.*

**Reviewer #2**

A small adaption of the overall structure could improve the readability of the paper further. Section 3 with the results contains with subsection 3.2 a comparison with MT_CKD. Section 4 is then about the comparison with laboratory observations. Although section 3.2 is about optical depth and section 4 mostly about cross sections, it could be an advantage to lift both to the same level of sections.

*We agree that the treatment of MT_CKD and the laboratory observations is inconsistent. Section 3 now exclusively refers to the Langley-derived optical depth, with all comparison taking place in individual sections; Section 4 is a short section detailing the comparison with MT_CKD (previously Section 3.2), and Section 5 now deals with the laboratory data comparison.*

Specific comments

P3, line 13: The authors state that "in many cases they use either version 2.5 or version3.2" of MT_CKD. It would be helpful, if the authors could give a more specific reference or a short indicative list of some relevant cases.

*We agree that including some examples with citations is beneficial here; we have included a list of various codes which use MT_CKD.*

*P3 L14 now says: Examples of codes using this model include the Atmospheric Radiative Transfer Simulator (Buehler et al. 2018), the Reference Forward Model (Dudhia et al. 2017), the Orbiting Carbon Observatory-2 (O'Dell et al 2018), the Met Office Unified Model (Walters et al. 2019) and the GFDL Global Atmosphere and Land Model (Zhao et al. 2018).*

P4, line 22: How do the authors come to the conclusion that Zugspitze measurements were taken at airmass factors of ~6? Please explain this in more detail.

*This number is taken from the fact that the lowest airmass factor at the surface for Zugspitze in Dec/Jan is ~3, and the limit imposed by the authors indicating that only observations with airmass below 9 are used. The sentence has been changed to reflect this range of airmasses more precisely.*

*P4 L23 now states: "To obtain a long enough path length to mitigate the lack of water vapour, the Zugspitze measurements were taken at large airmass factors (~3-9)."*

P5 concerning experimental setup:  It remains unclear how the Microtops II sunphotometer was operated.  Was this handheld device mounted on a stand/tripod?  Was it mounted on a solar tracker?  It would be helpful if the authors describe how it was ensured that the aerosol optical depth measurements were performed along the same atmospheric path.

*The Microtops sunphotometer was operated by hand by an operator at the field site co-incident with the FTIR measurements – the lack of a dedicated solar tracker on this is potentially a source of uncertainty in the aerosol measurements. However, this is not something that we can quantify or cross-check, given the lack of measurements from another source other than the FTIR. We have included this caveat when discussing the Microtops measurements, and added the need for better tracking of the solar disc as a suggestion in Section 5 (now 6).*

*P5 L29 now states "The Microtops has a field of view of 2.5°, and was operated by hand rather than mounted on a solar tracker, which could lead to some additional uncertainty (see Section 2.4)."*

*P37 L18 now states "Mounting our sunphotometer on a solar tracker may have aided our analysis and possibly reduced some of the problems described in Section 2.4."*

P9, fig. 3c: The data shown is marked as smoothed. How exactly was this smoothing mathematically performed? Is it the same smoothing about 15 cm-1 mentioned for the continuum on p6, line 14?

*This is the same smoothing (at 15 cm$^{-1}$) as performed for the final analysis, which was performed using a moving average (boxcar) filter. The text has been updated to reflect this.*

*P8 L10 now states explicitly "…smoothing using a 15 cm-1 boxcar filter…"*

P10, fig. 4:  In this figure the blue shading corresponds to k=1 and the cyan shading to k=2 uncertainty. There seems to be an envelope below and above the cyan shading that is colored again blue. If there is a physical meaning of it, could the authors please explain it?

*These lines are there to demarcate the edges of the uncertainty limits. Given that they caused ambiguity as to their meaning, they have been removed.*

P13, line 8: A suggestion for improvement is to mention the magnitude of the field-of-views of both the Microtops and the FTS.

*We now include the FOV of the Microtops and FTS within the manuscript, and point toward our discussion of the forward scattering/FOV issue on Page 13.*

*P5 L29 now states "The Microtops has a field of view of 2.5°, and was operated by hand rather than mounted on a solar tracker, which could lead to some additional uncertainty (see Section 2.4)."*

P16, fig. 8:  Is there any reason why the Langley and closure method derived optical depths in the upper part of the figure do not cover the same region of the residual in the lower part of the figure? If possible, they should be the same

*We have now updated Figure 8 to show the residual on the same scale as the top panel.*

.P19, line 4: The section title with laboratory observations fits to the lab measurements, but does not quite fit to the comparison with Reichert and Sussmann (2016) that are also included in the comparison.   Their observations were field observations as the CAVIAR field data in this paper.

*Section 4 (now Section 5, see top-level response to Reviewer #2 above) has been renamed to "Comparison with other observations", and the text in the first paragraph changed to reflect that we are also comparing to these field observations.*

*P19, L5 now reads "This section describes the relevant laboratory and field measurements…"*

P19, line 8: The derived continuum optical depth $\tau_{total}^{CAV}$ has another naming in the following formulas, e.g. formulas (3) and (5). Additionally, the quantity $\tau_{for}^{lab}$ mentioned on P20, fig. 11 was not introduced

*$\tau_{total}^{CAV}$ has been renamed to more accurately fit what is in the Figures and the Equations. Equation (4) has been updated to fix a typo, where the left-hand-side of the equation is equal to $\tau_{for}^{CAV}$ rather than the correct $\tau_{for}^{lab}$. We have also changed Figure 11 to now show the correct variables.*

.P24, line 10: The authors refer to lower temperature data (cyan point and dashed line), but in figure 13 there is no cyan point and no cyan dashed line. Seemingly, this passage is from an earlier version of this paper. CAVIAR-lab (297K) should be removed from the legend in figure 13.

*The versions of Figures 13 and 14 used in the uploaded drafts of the paper do not include this data point and the corresponding extrapolation – this was in error. The correct versions of these Figures are shown below – they are the figures uploaded for the original draft, but including the 297 K data in the legend. These are now included in the manuscript.*

[Figure]

P26, fig. 14: CAVIAR-lab (297) is not anymore included in the figure, so it should be removed from the legend. The same applies to the caption.

*See previous comment.*

P29, fig. 16: At the beginning of the second line of the caption self-continuum is assumed to be the foreign-continuum.

*The caption has been fixed to reflect that the CAVIAR-field self-continuum is estimated using the CAVIAR-lab foreign-continuum.*

*P29 L3 now reads: "Figure 16: Self-continuum from CAVIAR-field as estimated using (a) the MT_CKD foreign-continuum and (b) the CAVIAR-lab foreign-continuum, alongside MT_CKD_3.2 and selected laboratory measurements. The grey shaded regions indicate the k = 1 confidence limits in the CAVIAR-field self-continuum, and the blue shaded regions the uncertainty in the temperature-extrapolated (to 280 K) CAVIAR-lab data. The darker shaded regions are where these uncertainty limits overlap. The CAVIAR-lab uncertainties are obtained via Monte Carlo fits using the uncertainties in the higher-temperature ( > 350 K) CAVIAR-lab data." This also reflects the changes made in response to Reviewer #1's comment about the uncertainties in the CAVIAR-lab data.*

P33, fig. 17: In the caption it would be more precise, to mention that the showed data corresponds only to atmospheric windows in the mentioned region. The authors could insert "in atmospheric windows" between continuum and across.

*We now make explicit that the CAVIAR-field data corresponds to the window regions only, and use "CAVIAR-field" rather than "Langley-estimated" for consistency.*

*P33 L3 now says "CAVIAR-field foreign continuum in the atmospheric windows across the 2000-7000 cm-1 region…"*

P33, fig. 17: Concerning the showed Reichert and Sussmann (2016) data, ignored is the fact that they used MT_CKD_2.5.2 model for their continuum retrieval. As the self-continuum was assumed to be consistent with the MT_CKD model a direct comparison like in this figure is challenging.

*We believe that this comparison is reasonable, given we have used MT_CKD_3.2 in panel a) of this Figure to obtain the foreign continuum, and that since the foreign continuum contribution is dominant in the Reichert and Sussmann (2016) case, we believe that it is not likely to have an impactful effect on the comparison. We have added an additional sentence to the text to make clear that this is the case.*

*P30 L32 now says "Reichert and Sussmann indicate that the foreign continuum is by far the dominant contributor to the continuum in the majority of their measured spectral regions; we therefore compare their measurements to our foreign continuum measurements directly, but there may be some small self-continuum component which we do not account for in the Reichert and Sussmann data."*

P38, line 27:  Constraining the spectral coverage from 2000-7000 cm-1 to 2100-6600cm-1 would be more precisely .

*The text has been updated to match the title and more precisely reflect the wavenumber range.*

*P38 L29 now says "We have presented new field observations of the near-IR continuum in the atmospheric windows at 4, 2.1, 1.6, μm (2500-6600 cm-1)".*

Supplement: The airmass factor definition m = cos teta contradicts the airmass factor definition given in the paper.  The Beer-Bouguer-Lambert law given here is only valid with m = 1/cos teta.

*This was a typo and has been corrected.*

Technical corrections

P3, line 7: remove "at" after temperature

*This has been corrected to "as".*

P10, line 5: word repetition (distance 2) of approximation/approximately

*This sentence has been reworded to "…which in the limit of small absorption is approximately the optical depth noise in that region".*

P12, line 9: insert vapor (or vapour) between water and continuum

*This has been added.*

P27, line 3: Period/full stop is missing right after "term".

*This full stop has been added.*

P37, line 27, word repetition (distance 1) of aircraft

*We now reword this to remove the repetition, and also define FAAM.*

P40, line 16: remove "10", which seems to be a line number of an earlier version of this paper

*This has been removed.*

Supplement, P2: remove "the" in front of "account" in the third-to-last paragraph

*This has been removed.*

There is an inconsistency in the writing of the MT_CKD versions.  Mostly the current version is named MT_CKD3.2, but sometimes the naming is with a space in front of the version number. For the future reader a coherent way of writing would be an advantage, e.g. in browsing the paper. The model's developers are using with MT_CKD_3.2 a third way of spelling

*We now use the MT_CKD_ syntax, to be consistent with other literature.*

**Editor comments**

*We thank for the Editor for their comments, and agree with the requested changes. We go through these in turn below, along with our responses.*

p 23, lines 14+. The reasoning starting with "Interestingly" is misleading. The fact that the two field values based on different foreign continuum hypotheses are significantly different does not depend on the MT_CKD_3.2 self continuum. This inconsistency is foremost an indication of biases related to the two hypotheses. The conclusion that the assumption of a given foreign continuum is important for the actual analysis therefore is thus logically independent of the comparison with MT_CKD_3.2 and the disagreement with MT_CKD_3.2 should not be invoked to arrive at this conclusion.

Some of the following arguments in the paragraph might also be a little bit overstretched. Eg, the CAVIAR-lab 297 K data point seems to agree (within 1.2 sigma) with the Richard et al data and the CAVIAR lab temperature dependence. Consequently, there does not seem to be any anomaly here. Then, "the CAVIAR field estimate being closer to this temperature dependence when using the CAVIAR-lab foreign continuum", is not an argument, because it is based on the assumption that the Richard et al values must be used as a reference. If the Baranov & Lafferty data had been used as a reference then the CAVIAR-field w MT_CKD foreign continuum would be closer to that temperature dependence.

The whole paragraph (lines 8-22) needs to be rewritten as to avoid logical fallacies and circular reasoning.

*We agree that this paragraph needs rewriting, as it did not represent the argument we were attempting to make in the correct way. We have rewritten it to more clearly make the argument that the two estimates both seem to have some level of agreement with the two different implied temperature dependences, and neither agree very well with the MT_CKD_3.2 self-continuum, rather than the disagreement with MT_CKD_3.2 being evidence for the foreign continuum assumption being important.*

*We agree that Richard et al. is not a reference dataset, despite its high precision. Our conclusions have been made more conservative, mentioning that our measurements provide weak evidence for this lower temperature dependence, since there is better agreement with the estimate of the CAVIAR-field self-continuum with the real, experimentally observed foreign continuum from CAVIAR-lab, rather than the semi-empirical MT_CKD_3.2 foreign continuum.*

p 24, line 2 : set "k" in italics.

*This has been added.*

**Supplement,**

last phrase p1 : what is "gradient in which"

*Here we mean the gradient of the straight line as computed by ordinary least squares. We have reworded the sentence to make it clearer.*

3rd equation p2 : LHS (Delta tau ?) and equation number are missing

It seems that this equation should come after "Then subtracting tau' from tau:"

*This equation was intended to follow on from the previous one, as it is just the expansion of the bracketed terms in Eq. 13. We now make this explicit, and give it a unique specifier (Eq. 13b).*